# Bidirectional Convolutional Poisson Gamma Dynamical Systems

**Wenchao Chen**,* **Chaojie Wang**,* **Bo Chen**,† **Yicheng Liu**, **Hao Zhang**
National Laboratory of Radar Signal Processing
Xidian University, Xi'an, Shaanxi 710071, China
wcchen_xidian@163.com, xd_silly@163.com, bchen@mail.xidian.edu.cn,
mooooore66@gmail.com, zhanghao_xidian@163.com

**Mingyuan Zhou**
McCombs School of Business
The University of Texas at Austin
Austin, TX 78712, USA
mingyuan.Zhou@mccombs.utexas.edu

## Abstract

Incorporating the natural document-sentence-word structure into hierarchical Bayesian modeling, we propose convolutional Poisson gamma dynamical systems (PGDS) that introduce not only word-level probabilistic convolutions, but also sentence-level stochastic temporal transitions. With word-level convolutions capturing phrase-level topics and sentence-level transitions capturing how the topic usages evolve over consecutive sentences, we aggregate the topic proportions of all sentences of a document as its feature representation. To consider not only forward but also backward sentence-level information transmissions, we further develop a bidirectional convolutional PGDS to incorporate the full contextual information to represent each sentence. For efficient inference, we construct a convolutional-recurrent inference network, which provides both sentence-level and document-level representations, and introduce a hybrid Bayesian inference scheme combining stochastic-gradient MCMC and amortized variational inference. Experimental results on a variety of document corpora demonstrate that the proposed models can extract expressive multi-level latent representations, including interpretable phrase-level topics and sentence-level temporal transitions as well as discriminative document-level features, achieving state-of-the-art document categorization performance while being memory and computation efficient.

## 1 Introduction

How to represent documents to capture their underlying semantic structures is a key research problem in text analysis and language modeling [1–4]. It has been a long-standing challenge to capture long-range dependency in sequential data, especially for the word sequences of long documents that take discrete values for a large vocabulary. A simple remedy that helps capture long-range word dependency, but at the expense of sacrificing local structure, is to simplify the representation of each document as a bag of words (BoW) that ignores word order. Under this simplified BoW text representation, probabilistic topic models, such as latent Dirichlet allocation (LDA) [5, 6], Poisson

---

factor analysis (PFA) [7], and their various generalizations [7–11], have been widely deployed for text analysis, providing semantically meaningful latent representations. Vanilla topic models usually assume that each word is generated from a topic, characterized by a specific distribution over the terms in the vocabulary. Extracting a set of latent topics from a corpus, they characterize each document with its proportion over these topics. To enhance their modeling power and interpretability of vanilla topics models, which often have a single stochastic layer, a variety of deep generalizations with multiple stochastic layers have been proposed [12–19]. Despite providing improvement, their ultimate potentials have been limited by their use of BoW that completely ignores word local structure.

To remedy the loss of word order, Wang et al. [20] propose convolutional PFA (CPFA) as a convolutional generalization of PFA [7], extracting phrase-level (*i.e.*, $n$-gram) rather than word-level topics, where the number of words in each phrase is determined by the convolutional filter size that is typically set as $n = 3$. However, CPFA still ignores sentence order, limiting its power in capturing longer-range dependency beyond the filter size.

Treating text as word sequences, deep neural networks, such as convolutional neural networks (CNNs) [21–24] and recurrent neural networks (RNNs) [25–27], are wildly used to learn text representations [1, 28]. To efficiently capture long-distance relationships and obtain more expressive representations, there is a recent trend to construct a hierarchical language model [29–33] that models each document as a sequence of sentences, and each sentence as a sequence of words, achieving promising results in learning good document representations for down-stream tasks. For example, Tang et al. [31] employ CNNs/RNNs [34] to extract sentence representations, and summarize them into a document representation with another temporal architecture that models the sentence-level transitions. Yang et al. [32] introduce attention mechanism into the hierarchical language structure of Tang et al. [31] to further improve its classification performance. Moreover, the widely used Bidirectional Encoder Representations from Transformers (BERT) [35] has been extended in a hierarchical fashion to hierarchical BERT (HIBERT) for document summarization [29]. However, many of these deep neural network based methods need pretrain with a large amount of extra text data, followed by downstream task-specific finetune, to achieve good performance, and it is still often a challenge for these black-box methods to visualize and explain the semantic meanings learned by them.

Moving beyond the constraints of previous work, we represent each document as a sequence of sentences, each sentence as a sequence of word tokens, and each word token as a one-hit vector of dimension $V$, where $V$ is the vocabulary size. This provides a lossless representation of a corpus that respects the natural semantic structure of each document. Under such a lossless representation, we first propose convolutional Poisson gamma dynamical system (conv-PGDS) that adopts probabilistic convolutional structure [7, 20] to extract phrase-level features and models the inter-sentence dependency by a dynamical network. To exploit the sentence order, we further employ a bidirectional setting on our model and propose bidirectional conv-PGDS (bi-conv-PGDS). For scalable training and fast out-of-sample prediction, we integrate a stochastic gradient MCMC [36–40] and a convolutional-recurrent variational inference network to perform posterior inference. We further propose an attention bi-conv-PGDS (attn-bi-conv-PGDS) for supervised learning, combining the representation power of bi-conv-PGDS, discriminative power of deep neural networks, and selection power of the attention mechanism [32, 41, 42] under a principled probabilistic framework. On a variety of text corpora, we show the proposed models can not only capture long-distance relationships by exploiting the natural hierarchical structure inside each document, but also inherit various virtues of probabilistic topic models. The main contributions of this work are summarized as follows:

- Leveraging both word and sentence order, we propose conv-PGDS and its bidirectional generation, bi-conv-PGDS, to incorporate the document-sentence-word hierarchical structure into Bayesian hierarchical modeling. They capture both intra-sentence structure with convolution and inter-sentence dependency with gamma Markov chains. To the best of our knowledge, bi-conv-PGDS is the first unsupervised bidirectional hierarchical probabilistic model for document modeling.

- To achieve scalable training and fast testing, we perform Bayesian inference by combining stochastic gradient-MCMC (SG-MCMC) and convolutional-recurrent variational inference networks.

- We develop a supervised version of bi-conv-PGDS for document categorization and incorporate attention mechanism into it to further enhance its performance.

- Without requiring expensive pretrain on huge extra data, our models achieve state-of-the-art results, with low memory requirement and fast computation, in a variety of document modeling tasks.

## 2 Convolutional dynamical systems for document modeling

We first present conv-PGDS that integrates the convolutional components of CPFA [20] to model intra-sentence structure, dynamic components of PGDS [43] to model inter-sentence dependency, and a probabilistic pooling operation, which handles sentences of different lengths, into a well-defined generative model for document analysis. We then add a bidirectional generalization. To help better understand the hierarchical model, we summarize the notations in Table 4 in Appendix.

### 2.1 Convolutional Poisson gamma dynamical systems

Denote $\mathcal{D}_j = \{\boldsymbol{X}_{j1}, ..., \boldsymbol{X}_{jT_j}\}$ as the $j$th document with $T_j$ sentences and $\boldsymbol{X}_{jt} = [\boldsymbol{x}_{jt1}, ..., \boldsymbol{x}_{jtL_{jt}}]$ as its $t$th sentence of length $L_{jt}$, where $\boldsymbol{x}_{jtl}$ is the $l$th word token in this sentence represented with a one-hot vector as $\boldsymbol{x}_{jtl} = (x_{jtl1}, \dots, x_{jtlV})' \in \{0,1\}^V$, with $x_{jtlv} = 1$ if and only if token $l$ matches term $v$ of the vocabulary of size $V$. Distinct from CPFA that considers word order but ignores sentence order, we propose conv-PGDS that not only captures the internal temporal structures of each sentence using convolution, but also models the information transmissions between consecutive sentences using a gamma Markov chain. The generative process of conv-PGDS is expressed as

$$\boldsymbol{w}_{j1k} \sim \mathrm{Gam}\left(\tau_0 v_k, 1/\tau_0\right), \boldsymbol{w}_{jtk} \sim \mathrm{Gam}\left(\tau_0(\boldsymbol{\Pi}_{k:}\sum\nolimits_{s=1}^{S_{j,t-1}} \boldsymbol{w}_{j,t-1,:s}/S_{j,t-1})\mathbf{1}_{S_{jt}}, 1/\tau_0\right),$$
$$\boldsymbol{X}_{jt} = \mathbf{1}\left(\boldsymbol{M}_{jt} > 0\right), \boldsymbol{M}_{jt} \sim \mathrm{Pois}\left(\delta_{jt}\sum\nolimits_{k=1}^{K} \boldsymbol{D}_k * \boldsymbol{w}_{jtk}\right), \boldsymbol{D}_k\left(:\right) \sim \mathrm{Dir}\left(\eta\mathbf{1}_{VF}\right),$$
(1)

as described in detail below. Conv-PGDS factorizes each sentence $\boldsymbol{X}_{jt} = [\boldsymbol{x}_{jt1}, \dots, \boldsymbol{x}_{jtL_{jt}}] \in \{0,1\}^{V \times L_{jt}}$, represented with $L_{jt}$ sequential one-hot vectors, under the Bernoulli-Poisson link [44]. It defines $\boldsymbol{X}_{jt}$ by thresholding count matrix $\boldsymbol{M}_{jt} \in \{0,1,2,\dots\}^{V \times L_{jt}}$ that is factorized under the Poisson likelihood. It uses $\delta_{jt} > 0$ as a sentence-specific scaling factor and $\boldsymbol{D}_k = (\boldsymbol{d}_{k1}, \dots, \boldsymbol{d}_{kF}) \in \mathbb{R}_+^{V \times F}$ as the $k$th convolutional filter (*i.e.*, $F$-gram phrase-level topic), whose filter size is $F$ and vectorized form is $\boldsymbol{D}_k(:) = (\boldsymbol{d}'_{k1}, \dots, \boldsymbol{d}'_{kF})' \in \mathbb{R}_+^{VF}$. A Dirichlet prior is used to impose a simplex constraint on $\boldsymbol{D}_k(:)$. We denote the convolutional representation of sentence $t$ in document $j$ as $\boldsymbol{W}_{jt} = (\boldsymbol{w}_{jt1}, ..., \boldsymbol{w}_{jtK})' \in \mathbb{R}_+^{K \times S_{jt}}$, where $\boldsymbol{w}_{jtk} \in \mathbb{R}_+^{S_{jt}}$ is the feature map of sentence $t$ under topic $k$, and $S_{jt} := L_{jt} - F + 1$ varies with the sentence length $L_{jt}$. To capture the semantic or syntactic relations between the sentences in a document, which is often a challenging task [31], we assume $\boldsymbol{w}_{jt}$ to be gamma distributed and factorize its shape parameters using transition matrix $\boldsymbol{\Pi} \in \mathbb{R}_+^{K \times K}$, which captures cross-sentence temporal dependence, and the average pooling of the feature representation of sentence $t-1$, expressed as $\sum_{s=1}^{S_{j,t-1}} \boldsymbol{w}_{j,t-1,:s}/S_{j,t-1}$. We denote $\boldsymbol{\Pi}_{k:}$ as the $k$th row of $\boldsymbol{\Pi}$ and $\boldsymbol{w}_{j,t-1,:s}$ as the $s$th column of $\boldsymbol{W}_{j,t-1}$. Distinct from traditional deterministic feedforward pooling operations, which cut off the backward message passing, the probabilistic pooling layers in conv-PGDS can be trained jointly with the whole network, which enables the inter-sentence information transitions to influence $\{\boldsymbol{w}_{jt}\}_{t=1}^{T}$, the intra-sentence convolutional representations. Inspired by related temporal construction [43], which uses a (truncated) gamma process and the interactions of its atom weights to define a well-regularized transition matrix with tractable inference, we introduce $K$ factor weights $\boldsymbol{v} = (v_1, \dots, v_K)$ to model the strength of each component and let

$$\boldsymbol{\pi}_k \sim \mathrm{Dir}\left(v_1 v_k, \dots, \xi v_k, \dots, v_{K_l} v_k\right), \ v_k \sim \ \mathrm{Gam}\left(\gamma_0/K, 1/\beta\right), \quad (2)$$

where $\boldsymbol{\pi}_k = (\pi_{1k}, \dots, \pi_{Kk})'$ is the $k$th column of $\boldsymbol{\Pi}$ and $\pi_{k_1 k}$ can be interpreted as the strength of transitioning from topic $k$ to topic $k_1$. To complete the model, we let $\delta_{jt}, \xi, \beta \sim \mathrm{Gam}(\epsilon_0, 1/\epsilon_0)$.

Similar to CPFA [20], conv-PGDS extracts both global cooccurrence patterns and local temporal structures by forming $F$-gram phrase-level topics into $\boldsymbol{D}_k$, such as "how do you" and "microsoft email address" when $F = 3$. Distinct from CPFA, it extracts $\boldsymbol{\Pi}$ to model temporal dependence across sentences, capturing the transition patterns between the topics exhibited by consecutive sentences (*e.g.*, if "how do you" is an active topic in a sentence, then it is likely to activate its next sentence to exhibit topic "I am fine"). In conv-PGDS, embedded into the sentence-level feature $\boldsymbol{w}_{jt}$ are both phrase-level information of sentence $t$ and transition information of sentences 1 to $t-1$.

To summarize, conv-PGDS models both word order, by convolving over sequentially ordered one-hot word vectors, and sentence order, by adding dependency between the latent features of consecutive sentences. Note conv-PGDS reduces to CPFA [20] if ignoring its gamma Markov chain that models sentence order, and to PGDS [43] if removing both the convolution and pooling operations.

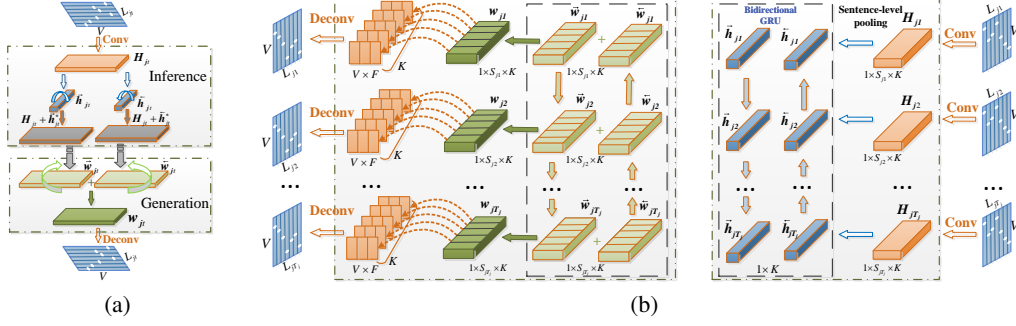

Figure 1: (a): Overview of bi-conv-PGDS with convolutional-recurrent variational inference network. (b): Detailed structure of bi-conv-PGDS (left) and convolutional-recurrent variational inference network (right).

## 2.2 Bidirectional convolutional Poisson gamma dynamical systems

We expand conv-PGDS in (1) to bidirectional conv-PGDS (bi-conv-PGDS), as shown in Fig. 1(b), which is a bidirectional dynamical model trained using all available input information in both the past and future of a specific time frame. The generative model of bi-conv-PGDS is expressed as

$$\overleftarrow{\boldsymbol{w}}_{jtk} \sim \text{Gam}\left(\tau_0(\overleftarrow{\boldsymbol{\Pi}}_{k:}\sum_{s=1}^{S_{j,t+1}}\overleftarrow{\boldsymbol{w}}_{j,t+1,:s}/S_{j,t+1})\mathbf{1}_{S_{jt}}, 1/\tau_0\right), \quad \overleftarrow{\boldsymbol{w}}_{jT_jk} \sim \text{Gam}\left(\tau_0\overleftarrow{v}_k, 1/\tau_0\right),$$

$$\vec{\boldsymbol{w}}_{jtk} \sim \text{Gam}\left(\tau_0(\vec{\boldsymbol{\Pi}}_{k:}\sum_{s=1}^{S_{j,t-1}}\vec{\boldsymbol{w}}_{j,t-1,:s}/S_{j,t-1})\mathbf{1}_{S_{jt}}, 1/\tau_0\right), \quad \vec{\boldsymbol{w}}_{j1k} \sim \text{Gam}\left(\tau_0\vec{v}_k, 1/\tau_0\right), \qquad (3)$$

$$\boldsymbol{X}_{jt} = 1\left(\boldsymbol{M}_{jt} > 0\right), \quad \boldsymbol{M}_{jt} \sim \text{Pois}\left(\delta_t\left(\sum_{k=1}^K \boldsymbol{D}_k * \left(\vec{\boldsymbol{w}}_{jtk} + \overleftarrow{\boldsymbol{w}}_{jtk}\right)\right)\right), \quad \boldsymbol{D}_k(:) \sim \text{Dir}\left(\eta\mathbf{1}_{VF}\right),$$

where $\vec{\boldsymbol{w}}_{jt}$ is the forward hidden feature map and $\overleftarrow{\boldsymbol{w}}_{jt}$ is the backward one. We let $\boldsymbol{w}_{jt} = \vec{\boldsymbol{w}}_{jt} + \overleftarrow{\boldsymbol{w}}_{jt}$ to summarize the neighboring sentences around sentence $t$. To interpret the bidirectional temporal relationship of the topics $\boldsymbol{D}_k$ in (3), we list the expected value of $\boldsymbol{M}_{jt}$ as

$$\mathbb{E}\left[\boldsymbol{M}_{jt}|\left\{\boldsymbol{D}, \vec{\boldsymbol{w}}_{j,t-1}, \overleftarrow{\boldsymbol{w}}_{j,t+1}, \vec{\boldsymbol{\Pi}}, \overleftarrow{\boldsymbol{\Pi}}\right\}\right] = \boldsymbol{D} * \left[\vec{\boldsymbol{\Pi}}\left(\left(\vec{\boldsymbol{w}}_{j,t-1,\cdot}/S_{j,t-1}\right)\mathbf{1}_{S_{jt}}\right)\right] + \boldsymbol{D} * \left[\overleftarrow{\boldsymbol{\Pi}}\left(\left(\overleftarrow{\boldsymbol{w}}_{j,t+1,\cdot}\Big/S_{j,t+1}\right)\mathbf{1}_{S_{jt}}\right)\right],$$

which shows that due to the bidirectional dynamic structure, the expected value of $\boldsymbol{M}_{jt}$ is divided into two parts: one from the forward feature representation, while the other from the backward one. In addition, $\vec{\boldsymbol{\Pi}}$ and $\overleftarrow{\boldsymbol{\Pi}}$ play the role of transiting the latent representations across sentences. Distinct from conv-PGDS, $\boldsymbol{w}_{jt}$ inferred by bi-conv-PGDS contains transition information of not only past sentences $1 \to t-1$, but also future ones $t+1 \leftarrow T_j$.

## 3 Bayesian inference and extensions to supervised learning

In this section, we first introduce a Gibbs sampling algorithm for bi-conv-PGDS and a corresponding stochastic gradient MCMC algorithm for scalable inference. To make our models both fast in out-of-sample prediction and easy to incorporate extra side information like document labels, we further construct a novel Weibull distribution based convolutional-recurrent variational inference network. In addition, we introduce a supervised learning component enhanced with attention modules, which can be jointly trained with bi-conv-PGDS using a hybrid SG-MCMC and variational inference.

### 3.1 Gibbs sampling

By generalizing the variable augmentation and marginalization techniques related to the Poisson, gamma, categorical, and Dirichlet distributions [7, 13], we propose a Gibbs sampling algorithm for bi-conv-PGDS, with all update equations and their derivations deferred to the Appendix. Here we note that the local variables of documents can be updated in parallel in each iteration, and the variable augmentation operations for sentence representation learning can also be parallelized at the sentence level. Thus the time cost of each Gibbs sampling iteration can be greatly reduced with Graphical Process Units (GPUs). While the computation of the Gibbs sampler can be accelerated inside each iteration, it requires processing all documents in each iteration and hence has limited scalability.

For scalable inference, we update our global parameters by generalizing TLASGR-MCMC [17, 18], a SG-MCMC algorithm that is developed to sample simplex-constrained global parameters in a mini-batch learning setting and provide high sampling efficiency by preconditioning its gradient with the Fisher information matrix. We defer the details to the Appendix.

## 3.2 Hybrid SG-MCMC and convolutional-recurrent autoencoding variational inference

Regardless of whether Gibbs sampling or TLASGR-MCMC is used, the need to perform a sampling based iterative procedure at the testing time limits the efficiency of the model for out-of-sample prediction. For the current model and inference, it is also difficult to incorporate the label information. As in Fig. 1(a), to make our model fast in out-of-sample prediction, we develop a convolutional-recurrent variational inference network to map the observation directly to its latent representations. Following Zhang et al. [18], the Weibull distribution is used to approximate the gamma distributed conditional posterior of $\vec{w}_{jt}, \overleftarrow{w}_{jt}$. As illustrated in Fig. 1(b), we first introduce an autoencoding variational distribution as $q(\{\vec{w}_{jt}, \overleftarrow{w}_{jt}\}_{j=1,t=1}^{N,T_j}) = \prod_{j=1}^{N}\prod_{t=1}^{T_j} q(\vec{w}_{jt}) \prod_{j=1}^{N}\prod_{t=T_j}^{1} q\left(\overleftarrow{w}_{jt_{jt}}\right)$, specifically

$$
\begin{aligned}
q\left(\vec{w}_{jtk}|-\right) &= \text{Weibull}\left(\vec{\Sigma}_{jtk} + (\vec{\Pi}_{k:}\sum_{s=1}^{S_{j,t-1}} \vec{w}_{j,t-1,:s}/S_{j,t-1})\mathbf{1}_{S_{jt}}, \vec{\Lambda}_{jtk}\right), \\
q\left(\overleftarrow{w}_{jtk}|-\right) &= \text{Weibull}\left(\overleftarrow{\Sigma}_{jtk} + (\overleftarrow{\Pi}_{k:}\sum_{s=1}^{S_{j,t+1}} \overleftarrow{w}_{j,t+1,:s}/S_{j,t+1})\mathbf{1}_{S_{jt}}, \overleftarrow{\Lambda}_{jtk}\right),
\end{aligned}
\tag{4}
$$

where $\vec{\Sigma}_{jt}, \overleftarrow{\Sigma}_{jt}, \vec{\Lambda}_{jt}, \overleftarrow{\Lambda}_{jt} \in {}^{K \times S_{jt}}$ are the parameters of $q(\vec{w}_{jt})$, $q(\overleftarrow{w}_{jt})$. To exploit both intra-sentence phrase-level structure and inter-sentence temporal dependencies, we generalize related constructions in Guo et al. [27, 45] to develop a convolutional-recurrent variational inference network, as shown in Fig. 1(b). The parameters above are deterministically transformed from the observation $\boldsymbol{X}_{jt}$ as

$$
\begin{aligned}
\mathbf{H}_{jt} &= \text{relu}\left(\mathbf{C}_1 * \mathbf{X}_{jt} + \mathbf{b}_1\right), \vec{h}_{jt} = \overrightarrow{\text{GRU}}\left(\text{pool}\left(\mathbf{H}_{jt}\right)\right), \overleftarrow{h}_{jt} = \overleftarrow{\text{GRU}}\left(\text{pool}\left(\mathbf{H}_{jt}\right)\right), \\
\vec{\Sigma}_{jt} &= \exp\left(\vec{C}_2 * \text{pad}\left(\mathbf{H}_{jt} + \vec{h}_{jt}^*\right) + \vec{b}_2\right), \overleftarrow{\Sigma}_{jt} = \exp\left(\overleftarrow{C}_2 * \text{pad}\left(\mathbf{H}_{jt} + \overleftarrow{h}_{jt}^*\right) + \overleftarrow{b}_2\right) \\
\vec{\Lambda}_{jt} &= \exp\left(\vec{C}_3 * \text{pad}\left(\mathbf{H}_{jt} + \vec{h}_{jt}^*\right) + \vec{b}_3\right), \overleftarrow{\Lambda}_{jt} = \exp\left(\overleftarrow{C}_3 * \text{pad}\left(\mathbf{H}_{jt} + \overleftarrow{h}_{jt}^*\right) + \overleftarrow{b}_3\right)
\end{aligned}
\tag{5}
$$

where $\boldsymbol{b}_1, \vec{b}_2, \overleftarrow{b}_2, \vec{b}_3, \overleftarrow{b}_3 \in \mathbb{R}^K$, $\boldsymbol{C}_1 \in \mathbb{R}^{K \times V \times F}$, $\vec{C}_2, \overleftarrow{C}_2, \vec{C}_3, \overleftarrow{C}_3 \in \mathbb{R}^{K \times K \times F}$. $\boldsymbol{H}_{jt} \in \mathbb{R}^{K \times S_{jt}}$, $\text{pad}(\cdot)$ denotes zero-padding, and $\text{pool}(\boldsymbol{H}_{jt}) \in \mathbb{R}^K$ denotes sentence-level average-pooling as defined before. We denote $\overrightarrow{\text{GRU}}$ and $\overleftarrow{\text{GRU}}$ are the forward and backward part of the bidirectional gated recurrent unit (GRU) [46] and $\vec{h}_{jt}, \overleftarrow{h}_{jt} \in \mathbb{R}^K$. With $\boldsymbol{H}_{jt} + \vec{h}_{jt}$ and $\boldsymbol{H}_{jt} + \overleftarrow{h}_{jt}$, realize by first broadcasting $\vec{h}_{jt}$ and $\overleftarrow{h}_{jt}$ to match the dimension of $\boldsymbol{H}_{jt}$, we fuse the intra-sentence structural information and inter-sentence temporal dependencies, enabling the proposed convolutional-recurrent variational inference network to learn rich latent representations for bi-conv-PGDS.

To train bi-conv-PGDS, we develop a hybrid SG-MCMC/autoencoding variational inference algorithm by combining TLASGR-MCMC and our proposed convolutional-recurrent variational inference network. Specifically, the global parameters $\{\boldsymbol{D}_k\}_{k=1}^{K}$, $\vec{\Pi}$ ,$\overleftarrow{\Pi}$ will be sampled with TLASGR-MCMC, while the parameters of the convolutional-recurrrent variational inference network, denoted by $\boldsymbol{\Omega}$, will be updated via stochastic gradient descent (SGD) by maximizing the evidence lower bound (ELBO) [47, 48]. The details are deferred to the Appendix.

## 3.3 Supervised attention bidirectional convolutional PGDS

To handle the document categorization task, we incorporate label supervision to extend bi-conv-PGDS into supervised bi-conv-PGDS (su-bi-conv-PGDS). To further improve its categorization performance, we combine it with the attention mechanism to develop supervised attention bi-conv-PGDS (attn-bi-conv-PGDS). Specifically, similar to Long et al. [49], to extract phrases important to the meaning of a sentence, we apply channel-wise attention in our latent feature representation of each sentence as

$$
\boldsymbol{u}_{jts} = \tanh\left(\boldsymbol{W}_w \boldsymbol{w}_{jt:s} + \boldsymbol{b}_w\right), \quad \alpha_{jts} = \frac{\exp(\boldsymbol{u}_{jts}^T \boldsymbol{u}_w)}{\sum_t \exp(\boldsymbol{u}_{jts}^T \boldsymbol{u}_w)}, \quad \boldsymbol{s}_{jt} = \sum_s \alpha_{jts} \boldsymbol{w}_{jt:s}
\tag{6}
$$

Table 1: Comparison of classification accuracy on unsupervisedly extracted features and runtime per iteration.

| Methods | Size | Accuracy | | | | Time in seconds | | | |
|---|---|---|---|---|---|---|---|---|---|
| | | Reuters | ELEC | IMDB-2 | IMDB-10 | Reuters | ELEC | IMDB-2 | IMDB-10 |
| LDA [5] | 1000 | 69.2±1.8 | 75.7±1.8 | 78.5±1.7 | 28.6±1.3 | 18.06 | 16.91 | 28.94 | 29.61 |
| DocNADE [50] | 1000 | 74.6±0.9 | 80.4±1.2 | 81.2±1.3 | 30.0±0.7 | 10.17 | 10.36 | 14.02 | 14.73 |
| DPFA [12] | 1000 | 71.1±1.8 | 77.2±1.9 | 78.8±1.9 | 28.6±1.0 | 40.34 | 38.51 | 56.87 | 60.25 |
| DPFA [12] | 1000-500 | 71.1±1.7 | 77.8±1.8 | 79.0±1.7 | 28.9±0.9 | 41.68 | 40.01 | 58.14 | 62.31 |
| DPFA [12] | 1000-500-200 | 71.2±1.7 | 78.4±1.6 | 79.3±1.7 | 29.0±0.9 | 43.19 | 41.25 | 59.02 | 64.03 |
| PGBN [13] | 1000 | 72.0±1.4 | 78.5±1.5 | 78.7±1.6 | 29.2±0.8 | 19.89 | 18.45 | 30.72 | 31.47 |
| PGBN [13] | 1000-500 | 73.7±1.3 | 80.1±1.5 | 80.9±1.4 | 30.1±0.6 | 29.84 | 25.83 | 43.01 | 45.24 |
| PGBN [13] | 1000-500-200 | 74.8±1.2 | 80.9±1.3 | 81.2±1.4 | 31.0±0.6 | 33.81 | 31.45 | 49.15 | 52.50 |
| WHAI [18] | 1000 | 71.2±1.6 | 78.0±1.7 | 78.9±1.7 | 28.9±0.9 | 9.61 | 9.71 | 12.35 | 13.02 |
| WHAI [18] | 1000-500 | 72.7±1.5 | 79.5±1.6 | 79.8±1.5 | 29.7±0.9 | 12.23 | 12.58 | 17.03 | 17.69 |
| WHAI [18] | 1000-500-200 | 73.9±1.3 | 80.1±1.6 | 80.4±1.4 | 30.4±1.0 | 14.12 | 14.71 | 20.51 | 21.49 |
| DPGDS [20] | 1000 | 73.6±1.0 | 79.2±0.9 | 80.2±0.8 | 31.2±0.6 | 26.34 | 23.58 | 41.26 | 43.08 |
| DPGDS [20] | 1000-500 | 74.5±1.0 | 79.9±0.8 | 81.0±0.8 | 32.0±0.6 | 39.62 | 34.83 | 60.11 | 64.22 |
| DPGDS [20] | 1000-500-200 | 75.4±0.9 | 81.4±0.8 | 81.7±0.8 | 32.5±0.5 | 47.75 | 41.62 | 71.86 | 77.64 |
| CPGBN [20] | 1000 | 75.6±0.8 | 81.4±0.9 | 81.8±0.9 | 34.2±0.5 | 20.10 | 18.76 | 30.92 | 31.68 |
| CPGBN [20] | 1000-500 | 76.9±0.7 | 82.1±0.9 | 82.3±0.8 | 35.0±0.6 | 39.86 | 34.59 | 60.19 | 65.22 |
| CPGBN [20] | 1000-500-200 | 77.3±0.7 | 82.5±0.8 | 82.6±0.7 | 35.3±0.5 | 47.54 | 41.28 | 71.31 | 76.86 |
| conv-PGDS | 1000 | 77.4±0.8 | 83.1±0.9 | 83.3±0.9 | 37.2±0.4 | 27.50 | 24.09 | 41.47 | 43.21 |
| bi-conv-PGDS | 1000 | **78.0±0.7** | **84.5±0.8** | **84.0±0.8** | **37.9±0.4** | 28.99 | 25.16 | 43.41 | 45.80 |

To reward the sentences that provide important clues to correctly classify a document, we again use attention mechanism to measure the importance of different sentences. This yields

$$\boldsymbol{u}_{jt} = \tanh\left(\boldsymbol{W}_s \boldsymbol{s}_{jt} + \boldsymbol{b}_s\right), \quad \alpha_{jt} = \frac{\exp(\boldsymbol{u}_{jt}^T \boldsymbol{u}_s)}{\sum_t \exp(\boldsymbol{u}_{jt}^T \boldsymbol{u}_s)}, \quad \boldsymbol{t}_j = \sum_t \alpha_{jt} \boldsymbol{s}_{jt}, \quad (7)$$

where $\boldsymbol{t}_j$ is a document-level feature vector that summarizes all the sentences of a document. Setting the label probability vector as $\boldsymbol{p}_j = (p_1, ..., p_C) = \text{softmax}(\boldsymbol{W}_c \boldsymbol{t}_j + \boldsymbol{b}_c)$, we adopt the cross-entropy loss between the ground truth label distribution $\boldsymbol{p}_j^g$ and predicted label distribution $\boldsymbol{p}_j$ as the training loss: $L_g = -\sum_{j=1}^M \sum_{c=1}^C p_{jc}^g \cdot \log(p_{jc})$. Thus the loss function of the entire framework is modified as $L = -L_g + \lambda L_s$, where $L_g$ refers to the negative ELBO of the generative model, and $\lambda > 0$ is a regularization hyper-parameter to balance data generation and label supervision and we set $\lambda = 0.1$ in our experiment.

## 4  Experiments

We evaluate the effectiveness of our model on five large corpora, including ELEC, IMDB-2, Reuters, Yelp 2014, and IMDB-10, which are described in detail in the Appendix. We fix the hyperparameters of our models as $\tau_0 = 1, \epsilon_0 = 0.1, \gamma_0 = 0.1, \eta = 0.05$ for all experiments. Python (PyTorch) code is provided at https://github.com/BoChenGroup/BCPGDS.

### 4.1  Unsupervised models

We first evaluate various models by extracting document representations in an unsupervised manner and training a linear classifier on these unsupervisedly extracted document features. We compare to a wide variety of topic models, including LDA [5], DocNADE [50], DPFA [12], PGBN [13], WHAI [18] , DPGDS [51], and CPGBN [20], where only CPGBN takes one-hot word sequences as its input, while all the others directly operate on BoW vectors that ignore word order. To make a fair comparison, we adopt comparable multilayer structures across all methods and set the same convolutional filter width $F = 3$ for both CPGBN and our models.

We list in Table 3 the network structures of various methods and their corresponding results, which are either quoted from the original papers or reproduced with the code provided by the authors. Given the same network structure, models with BoW input are shown to underperform those with one-hot vector input, suggesting the importance of the convolutional operation that helps leverage the word order information. Conv-PGDS and bi-conv-PGDS both provide clear improvement over CPGBN, suggesting the importance of capturing inter-sentence dependencies. Incorporating the contextual information into sentence representations that are then aggregated, bi-conv-PGDS extracts more expressive document-level features, as suggested by its consistent improvement over conv-PGDS and all basline methods. For computational complexity, we also report the average run time of each iteration

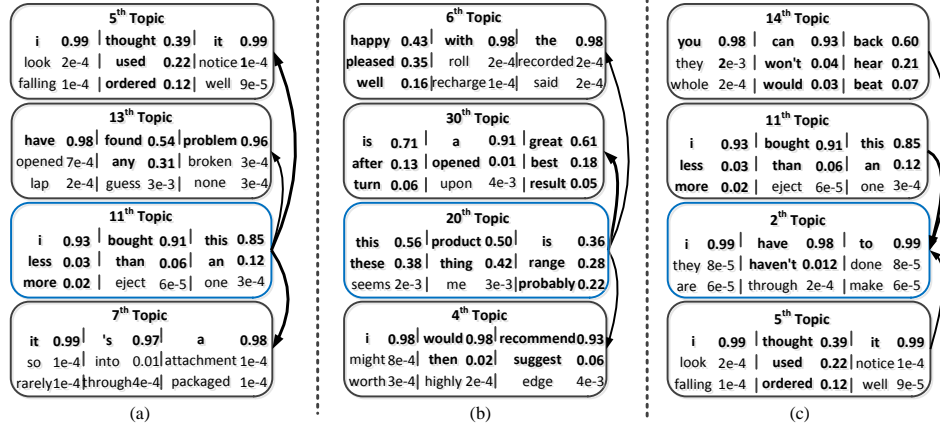

(a)        (b)        (c)

Figure 2: Three example transition relations between topics from ELEC learned by conv-PGDS (better understood together with Fig. 3). Representative phrases are highlighted in bold.

for all methods. Handling each document parallelly and accelerating Gibbs sampling procedure with GPU, our models outperform other non-dynamic models with a comparable, or even less, computational burden. In addition to quantitative evaluation, we also visualize the inferred convolutional filters and transition matrices of conv-PGDS, showing the advantages of the proposed Bayesian models over existing "black-box" deep neural networks in terms of having interpretable latent structures.

As shown in Fig .3, from the full transition matrix $\Pi$ learned from ELEC, we visualize the transitions between the 15 most frequently used convolutional filters. We visualize some example filters and how they transit to each other in Fig .2. We provide in the Appendix the visualizations of each of these 15 filters. For each column of convolutional filter $\boldsymbol{D}_k \in \mathbb{R}_+^{V \times 3}$, the top three terms that have the highest weights and their corresponding weights are exhibited. It is particularly interesting to notice that the top terms at different columns can be combined into a variety of interpretable phrases with related semantics. Taking the 11th filter as an example, its corresponding phrase-level topic and transition relationships are visualized in Fig .2 (a), which shows that filter 11 on "i bought this/an" is more likely to activate a transition to filter 5 on "i thought/used/ordered it", filter 7 on "it 's a", and filter 14 on "have found problem". In addition, as shown in Fig .2 (b), filter 20 on "this/these product/thing is/range" is the phrase describing "product" and it is more likely to activate a transition to filters 4, 6, 30 that are mainly about " product evaluations". The transition links shown in Fig .2 (a) and (c) correspond to column 11 and row 2 in Fig. 3, respectively. Capturing phrase-level topics and inter-sentence information transitions, the document representations of our models are embedded with rich semantic information, making them well suited for downstream tasks.

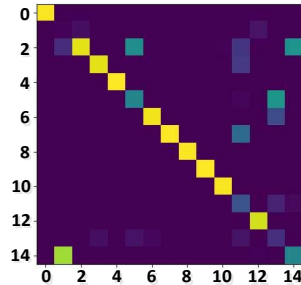

Figure 3: Transition matrix $\Pi$ for top 15 topics from ELEC learned by conv-PGDS.

To verify the efficiency of our generative model in capturing the word order information, we estimate the likelihood of a sentence with shuffled word order. Fig. 4 (left) shows the likelihood decreases as the shuffling rate increases, indicating CPGDS provides a higher confidence on real sentences than orderless ones. Moreover, we further estimate the likelihood of document with shuffled sentence order and observe similar behaviors in Fig. 4 (right), which illustrate that our generative model can also capture the sentence order information in a document.

## 4.2 Supervised models

To evaluate the effectiveness of attn-bi-conv-PGDS, which incorporates the label information and attention mechanism into bi-conv-PGDS, we compare it with various supervised methods on four popular benchmarks and list the results in Table 2. For comparison, we include BoW-based methods: sAVITM [52], MedLDA [10], sWHAI [18]; bag-of-n-gram models [53]: SVM-unigrams, SVM-bigrams, SVM-trigrams; CNN/RNN based methods: SVM-wv [54], LSTM-wv [55], CNN-wv [22], CNN-one-hot [56]; methods leveraging the document-sentence-word hierarchical structure: Conv-GRNN [31], LSTM-GRNN[31], HAN-AVE, HAN-ATT [32]; and convolutional probabilistic model: sCPGBN

Table 2: Comparison of classification accuracy and number of model parameters between supervised models

| Methods | Model size | ELEC | IMDB-2 | IMDB-10 | yelp14 | #Param |
|---|---|---|---|---|---|---|
| sAVITM [52] | 200 | 83.7 | 84.9 | 30.1 | 50.4 | 4.10M |
| MedLDA [10] | 200 | 84.6 | 85.7 | 30.7 | 51.0 | 2.10M |
| sWHAI-layer1 [18] | 200 | 86.8 | 87.2 | 32.0 | 52.7 | 4.10M |
| sWHAI-layer2 [18] | 200-100 | 87.5 | 88.0 | 33.5 | 53.4 | 4.14M |
| sWHAI-layer3 [18] | 200-100-50 | 87.8 | 88.2 | 34.2 | 54.3 | 4.15M |
| SVM-unigrams [53] | - | 86.3 | 87.7 | 39.9 | 58.9 | - |
| SVM-bigrams [53] | - | 87.2 | 88.2 | 40.9 | 57.6 | - |
| SVM-trigrams [53] | - | 87.4 | 88.5 | 41.3 | 59.8 | - |
| SVM-wv [54] | - | 85.9 | 86.5 | 42.4 | 59.6 | - |
| LSTM-wv [55] | 200 | 88.3 | 89.0 | 43.6 | 62.5 | 5.30M |
| CNN-wv [22] | 200 | 88.6 | 89.5 | 42.5 | 59.7 | 4.94M |
| CNN-one-hot [56] | 200 | 91.3 | 91.6 | 43.2 | 62.4 | 6.08M |
| LSTM-one-hot [55] | 200 | 91.6 | 91.8 | 44.1 | 63.2 | 8.16M |
| Conv-GRNN [31] | 200-100 | 91.4 | 91.5 | 44.5 | 63.7 | 4.94M |
| HAN-AVE [32] | 200-100 | 91.5 | 91.6 | 47.8 | 67.0 | 5.56M |
| HAN-ATT [32] | 200-100 | 91.7 | 91.8 | 49.4 | 68.2 | 5.68M |
| sCPGBN-layer1 [20] | 200 | 91.6±0.3 | 91.8±0.3 | 43.9±0.3 | 63.1±0.3 | 12.00M |
| sCPGBN-layer2 [20] | 200-100 | 92.0±0.2 | 92.4±0.2 | 45.0±0.3 | 64.2±0.3 | 12.04M |
| sCPGBN-layer3 [20] | 200-100-50 | 92.2±0.2 | 92.6±0.2 | 45.8±0.3 | 64.9±0.3 | 12.05M |
| DocBERT [57] | - | **93.4** | **94.1** | **54.2** | **72.0** | 110.0M |
| sconv-PGDS | 200 | 91.9±0.2 | 92.0±0.2 | 48.9±0.2 | 68.0±0.3 | 12.20M |
| su-bi-conv-PGDS | 200 | 92.5±0.2 | 92.6±0.2 | 51.2±0.2 | 70.5±0.3 | 12.24M |
| attn-bi-conv-PGDS | 200 | **93.0**±0.3 | **93.4**±0.3 | **53.8**±0.2 | **71.2**±0.3 | 12.28M |

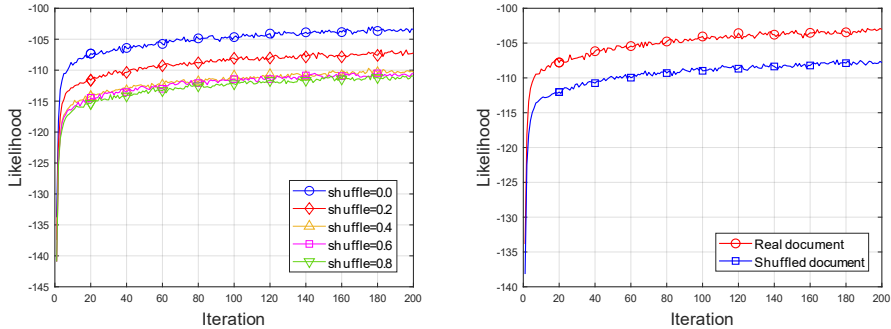

Figure 4: left: likelihood of shuffled sentence; right: likelihood of shuffled document

[20] and Transformer based pre-trained language models: DocBERT [57]. As shown in Table 2, BoW-based methods in general underperform the methods that take word order into consideration. Traditional methods based on hand-crafted features like SVM-unigrams/bigrams/trigrams typically underperform CNN/RNN based methods, such as CNN-wv, LSTM-wv, that learn local dependence structure from the data. CNN/RNN based methods, which do not exploit the hierarchical document structure, clearly underperform the methods that exploit these structures to capture long-term temporal dependence and cross-sentence information transmission, especially for large-scale datasets with many categories. Pretrained on huge extra text data, DocBERT with large memory requirement and high computational complexity recently achieves state-of-the-art document classification performance.

By incorporating hierarchical document structure into probabilistic generative models and the attention mechanism, attn-bi-conv-PGDS achieves comparable results with DocBERT but with significantly fewer parameters (about 90% reduction) and faster test speed (more than 92% reduction in testing time, as shown in Table 2.). Moreover, in order to validate that our model is able to attend to informative sentences and phrases in a document given the context, we provide two example visualizations of the hierarchical attention of our model and compare them with CPFA equipped with attention in Fig. 5. We select three most prominent sentences in a document for visualization.

| Dataset | BERT | attn-bi-conv-PGDS |
|---|---|---|
| ELEC | 145.2 | 8.1(94% ↓) |
| IMDB-2 | 175.1 | 13.0(93% ↓) |
| IMDB-10 | 241.3 | 19.0(92% ↓) |
| Yelp'14 | 1810.6 | 102.3(94% ↓) |

Table 3: Comparison of testing times (seconds) with batch-size 128 on two RTX 2080 Ti GPUs.

Blue and red denote the phrase weight and sentence weight, respectively. Darker color represents greater weight (the sentence weight is listed on its left-side). Due to the hierarchical structure, we normalize the word weight by the sentence weight to make sure that only important words in important sentence are emphasized. As we can see, attn-CPFA can only select the phrases carrying strong sentiments, but cannot deal with complex across-sentence context. Taking the document in Fig. 5 (b) as an example, attn-CPFA only focuses on phrase "the worst movie", thus to classify it into

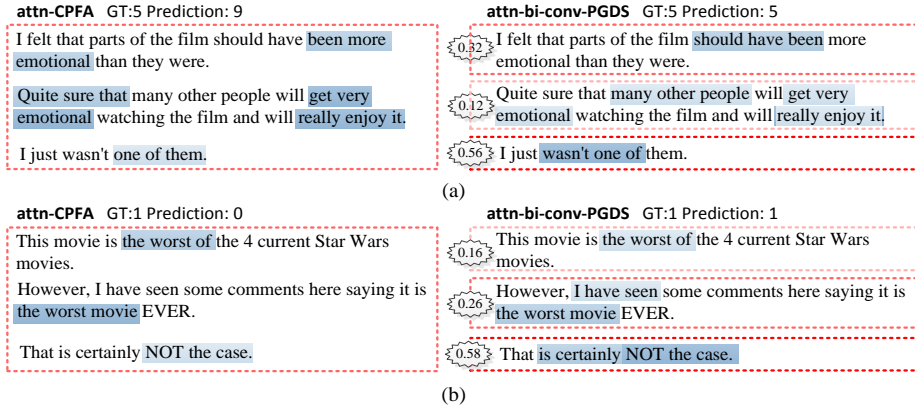

Figure 5: Two examples of attention from IMDB-10 (a) and IMDB-2 (b) learned by sCPFA and attn-bi-conv-PGDS.

the "negative" class. Our model captures the transition between sentences and deduces it as a positive review given its higher attention weight placed on the last sentence.

Our recurrent-convolutional variational autoencoder can be trained in a semi-supervised setting by modifying our supervised loss into $L^{semi} = \sum_{d \in \{D_l + D_u\}} L_g(d) + \xi \sum_{d \in D_l} L_s(d)$, where $D_l$ denotes the set of labeled data and $D_u$ unlabeled data. In this way, our attn-bi-conv-PGDS can improve upon supervised natural language tasks by leveraging features learned from unlabeled documents. In principle, a good unsupervised feature extractor will improve the generalization ability in semi-supervised learning setting. To demonstrate the advantage of incorporating the reconstruction objective into the training of document classifier, we evaluate our model with different amount of labeled data (5%, 10%, 25%, or 50%), besides the whole training set being unlabeled data.

We compare our model with LSTM-one-hot and HAN-ATT, which are purely supervised models, sCPGBN, which is a VAE structured model, and DocBERT, which can be pre-trained with unlabeled data using Masked LM (MLM) task. More details about semi-supervised classification with attn-bi-conv-PGDS and DocBERT are deferred to Appendix. From the result shown in Fig. 6, VAE structured models have better generalization performance varying with the size of the labeled training data, benefiting from naturally leveraging the unlabeled documents. It is interesting to notice that attn-bi-conv-PGDS outperforms DocBERT when the labeled training data becomes limited. We attribute the superior generalization performance of our model to the good representation power of bi-conv-PGDS, which serves as the decoder of our VAE framework.

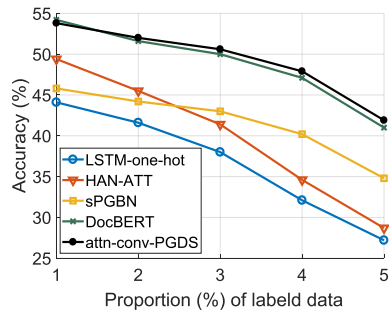

Figure 6: Semi-supervised classification accuracy on IMDB-10.

## 5 Conclusion

Respecting the natural hierarchical structure of each document (words forms a sentence, sentences form a documents) and incorporating that information into hierarchical Bayesian modeling, we propose convolutional Poisson gamma dynamical systems (conv-PGDS) to capture not only phrase-level topics, by introducing word-level convolutions, but also how the topic usages evolve over consecutive sentences, by modeling sentence-level transitions. To consider both forward and backward sentence-level information transmissions, we further develop a bidirectional conv-PGDS. For scalable inference, we develop a hybrid Bayesian inference algorithm that integrates SG-MCMC and a convolutional-recurrent variational inference network to approximate the posterior. In addition, we incorporate attention mechanism into our model and develop an attention guided bidirectional conv-PGDS for document categorization. Experiments on unsupervised feature extraction and supervised and semi-supervised document categorization tasks show that our models can not only extract discriminative document representations, achieving state-of-the-art results with low memory and computational cost, but also inherit the virtues of probabilistic topic models that provide interpretable phrase-level topics and sentence level-temporal transitions.

## Broader Impact

The proposed models are probabilistic topic models considering hierarchical temporal information within each document, which can not only achieve state-of-the-art performance in some text analysis tasks, such as unsupervised feature extraction and (semi-)supervised document categorization, but also obtain semantically meaningful topics and latent features as well as interpretable transition relations. Thus, they may be used to detect harmful articles which may contain fake news, violent content, and fraudulent materials. In addition, they can be used to recommend articles with specific contents to users according to their needs. More importantly, as they are able to provide interpretable latent features, one could try to understand why a certain categorization and recommendation has been made by the proposed model for a given article, so more appropriate actions can be taken rather than purely trusting the model itself to make the right decisions.

Although big pre-trained language models, such as BERT [35], GPT2 [58], and GPT3 [59], could be fine-tuned for a variety of natural language processing tasks to achieve state-of-the-art performance in many difference settings, they are consuming significant amount of computing resources, leading to higher energy consumption and $CO_2$ emissions. Comparing with these pretrained big models, the proposed probabilistic models provide customized probabilistic solutions to specific problems, achieving comparable results in the specific task of document categorization but with much lower memory and computational cost, which is beneficial for energy saving and environmental protection.

Providing good performance while maintaining interpretability becomes an even more urgent issue today given the recent trend in building larger and more complex black-box models trained with bigger data, which work well but make it become increasingly more difficult to understand how and why they work well. We hope our work can motivate machine learners to pay more attention to the study of interpretable and compact models. When evaluating the model, we should analyze the model more and find what can help different areas of society. At present, many people tend to strongly emphasize on the numerical performance and pay less attention to the energy cost of training and deploying these models. Meanwhile, the black box feature of deep learning means that the model may fail without clear explanations, so it is difficult to deploy them to applications with high-level safety and stability requirements. An interpretable model, on the other hand, enables people to understand what the model really learns and how it makes decisions, so as to better evaluate the model and understand how and where to deploy it for the benefits of the society.

## Acknowledgements

The authors thank the anonymous reviewers for their constructive comments and suggestions that have helped to improve the paper. B. Chen acknowledges the support of the Oversea Talent by Chinese Central Government, the 111 Project (No. B18039), NSFC (61771361), Shanxi Innovation Team Project, and the Innovation Fund of Xidian University. M. Zhou acknowledges the support of the U.S. National Science Foundation under Grant IIS-1812699.

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
