[Supplementary Material]

# A Gibbs Sampling for bi-conv-PGDS

It is a non-trivial task to develop Gibbs sampling update equations for the bi-conv-PGDS model, mainly due to the difficult to sample the gamma shape parameters from their conditional posteriors. By exploiting related variable augmentation and marginalization techniques of Zhou el al.[11] and their generalizations into the inference for gamma Markov chains [43, 51, 60], we propose a bidirectional Gibbs sampler to make it simple to compute the conditional posterior of the model parameters. We repeatedly exploit the following three properties, as summarized in [43], in order to do the inference.

## A.1 The notation table for our generative model

Table 4: Notation table for our generative model.

| symbol | meaning | symbol | meaning |
|---|---|---|---|
| $\mathcal{D}_j$ | $j$th input document | $\boldsymbol{X}_{jt}$ | $t$th sentence of $\mathcal{D}_j$ |
| $\boldsymbol{W}_{jt}$ | convolutional representation of $\boldsymbol{X}_{jt}$ | $\boldsymbol{w}_{jtk}$ | $k$th convolutional representation of $\boldsymbol{W}_{jt}$ |
| $\boldsymbol{\Pi}$ | Transition matrix | $\vec{\boldsymbol{w}}_{jtk}, \overleftarrow{\boldsymbol{w}}_{jtk}$ | forward, backward convolutional representation |
| $\delta_{jt}$ | sentence-specific scaling factor | $\boldsymbol{v}$ | factor weight |
| $\boldsymbol{D}_k$ | $k$th convolutional filter | $\boldsymbol{M}_{jt}$ | thresholding count matrix of $\boldsymbol{X}_{jt}$ |
| $\vec{\boldsymbol{\Pi}}, \overleftarrow{\boldsymbol{\Pi}}$ | forward, backward Transition matrices | - | - |

## A.2 Properties

**Property 1 (P1):** If $x. = \sum_{n=1}^{N} x_n$, where $x_n \sim \text{Poisson}(\theta_n)$ are independent Poisson-distributed random variables, then $(x_1, \ldots, x_N) \sim \text{Multinomial}\left(x, \frac{\theta_1}{\sum_{n=1}^{N} \theta_n}, \ldots, \frac{\theta_N}{\sum_{n=1}^{N} \theta_n}\right)$ and $x. \sim \text{Poisson}\left(\sum_{n=1}^{N} \theta_n\right)$ [7, 61]

**Property 2 (P2):** If $x \sim \text{Poisson}(c\theta)$, where $c$ is a constant, and $\theta \sim \text{Gam}(a, 1/b)$, then $x \sim \text{NB}\left(a, 1 - \frac{1}{1+bc}\right)$ is a negative binomial (NB) distributed random variable. We can equivalently parameterize it as $x \sim \text{NB}(a, g(\zeta))$, where $g(z) = 1 - \exp(-z)$ is the Bernoulli-Poisson link [44] and $\zeta = \ln(1 + bc)$.

**Property 3 (P3):** If $x \sim \text{NB}(a, g(\zeta))$ and $l \sim \text{CRT}(x, a)$ is a Chinese restaurant table (CRT) distributed random variable, then $x$ and $l$ are equivalently jointly distributed as $x \sim \text{SumLog}(l, g(\zeta))$ and $l \sim \text{Poisson}(a\zeta)$ [11]. The sum logarithmic (SumLog) distribution is further defined as the sum of $l$ independent and identically logarithmic-distributed random variables, i.e., $x = \sum_{i=1}^{l} y_i$ and $y_i \sim \text{Logarithmic}(g(\zeta))$.

## A.3 Inference

Similar to Wang et al. [20], to avoid directly process sparse document matrix, which will bring unnecessary burden in computation and storage, we apply variable augmentation under the Poisson likelihood [7, 13] to upward propagate latent count matrices $\boldsymbol{M}_{jt}$ as

$$(\boldsymbol{M}_{jt1}, \ldots, \boldsymbol{M}_{jtK} \,|\, -) \sim \text{Multi}\left(\boldsymbol{M}_{jt}; \frac{\boldsymbol{D}_1 * \boldsymbol{w}_{jt1}}{\sum_{k=1}^{K} \boldsymbol{D}_k * \boldsymbol{w}_{jtk}}, \ldots, \frac{\boldsymbol{D}_K * \boldsymbol{w}_{jtK}}{\sum_{k=1}^{K} \boldsymbol{D}_k * \boldsymbol{w}_{jtk}}\right) \quad (8)$$

where $\boldsymbol{w}_{jtk} := \vec{\boldsymbol{w}}_{jtk} + \overleftarrow{\boldsymbol{w}}_{jtk}$. Note only nonzero elements of $\boldsymbol{M}_{jtk} \in Z^{V \times L_{jt}}$ need to be focused on. By expanding the convolutional operation along the dimension of $\boldsymbol{D}_k$, we rewrite the likelihood function of $\boldsymbol{w}_{jtk}$ as

$$m_{jtkvl} \sim \text{Pois}\left(\delta_{jt} \sum_{f=1}^{F} d_{kvf} w_{jtk(l-f+1)}\right) \quad (9)$$

where $w_{jtk(l-f+1)} := 0$ if $l - f + 1 \notin \{1, 2, \ldots, S_{jt}\}$. Thus each nonzero element $m_{jtkvl}$ could be augment as

$$(d_{jtkvl1}, \ldots, d_{jtkvlF} \mid -) \sim \text{Multi}(m_{jtkvl}; \frac{d_{kv1}w_{jtk(l-1+1)}}{\sum_{f=1}^{F} d_{kv1}w_{jtk(l-1+1)}}, \ldots, \frac{d_{kvF}w_{jtk(l-F+1)}}{\sum_{f=1}^{F} d_{kvF}w_{jtk(l-F+1)}}) \tag{10}$$

Assigning $\boldsymbol{d}_{jtkvl} = (d_{jtkvl1}, \ldots, d_{jtkvlF}) \in \mathbb{Z}^F$. Via **P1**, we have

$$((\boldsymbol{d}'_{j \cdot k1 \cdot}, \ldots, \boldsymbol{d}'_{j \cdot kV \cdot})' \mid m_{j \cdot k \cdot \cdot}) \sim \text{Multi}(m_{j \cdot k \cdot \cdot}; \boldsymbol{D}_k(:)) \tag{11}$$

where $\cdot$ denotes summing over the corresponding index and $\boldsymbol{d}_{j \cdot kv \cdot} = \sum_{t=1}^{T_j} \sum_{l=1}^{L_{jt}} \boldsymbol{d}_{jtkvl}$, $m_{j \cdot k \cdot \cdot} = \sum_{t}^{T_j=1} \sum_{v=1}^{V} \sum_{l=1}^{L_{jt}} m_{jtkvl}$.

**Sampling the convolutional filter:** With the Dirichlet-multinomial conjugacy, we have

$$(\boldsymbol{D}_k(:) \mid -) \sim \text{Dir}((\boldsymbol{d}'_{\cdot k \cdot 1 \cdot}, \ldots, \boldsymbol{d}'_{\cdot k \cdot V \cdot})' + \eta \boldsymbol{1}_{VF}) \tag{12}$$

Similarly, we can expand the convolution along the dimension of $\boldsymbol{w}_{jtk}$ as $m_{jtkvl} \sim \text{Pois}(\sum_{s=1}^{S_{jt}} w_{jtks}d_{kv(l-s+1)})$, where $d_{kv(l-s+1)} := 0$ if $l - s + 1 \notin \{1, 2, \ldots, F\}$ and augment $m_{jtkvl}$ as

$$(m_{jtkvl1}, \ldots, m_{jtkvlS_{jt}} \mid -) \sim \text{Multi}\left(m_{jtkvl}; \frac{w_{jk1}d_{kv(l-1+1)}}{\sum_{s=1}^{S_{jt}} w_{jks}d_{kv(l-s+1)}}, \ldots, \frac{w_{jkS_{jt}}d_{kv(l-S_{jt}+1)}}{\sum_{s=1}^{S_{jt}} w_{jks}d_{kv(l-s+1)}}\right) \tag{13}$$

Assigning $\boldsymbol{m}_{jtkvl} = (m_{jtkvl1}, \ldots, m_{jtkvlS_{jt}}) \in \mathbb{Z}^{S_{jt}}$. By marginalizing out $\boldsymbol{D}_k$, we can use **P1** to decouple $\boldsymbol{D}_k * \boldsymbol{w}_{jtk}$ and get

$$\boldsymbol{m}_{jtk \cdot \cdot} \sim \text{Pois}(\delta_{jt}\boldsymbol{w}_{jtk}) \tag{14}$$

Assigning $\boldsymbol{A}_{jtk} = \sum_{v=1}^{V} \sum_{l=1}^{L_{jt}} \boldsymbol{m}_{jtkvl}$. As $\boldsymbol{w}_{jtk} = \vec{\boldsymbol{w}}_{jtk} + \overleftarrow{\boldsymbol{w}}_{jtk}$, via **P1**, we can separate $\vec{\boldsymbol{w}}_{jtk}$ and $\overleftarrow{\boldsymbol{w}}_{jtk}$ by augmenting $\boldsymbol{A}_{jtk}$ as $\boldsymbol{A}_{jtk} = \vec{\boldsymbol{A}}_{jtk} + \overleftarrow{\boldsymbol{A}}_{jtk}$, where

$$\vec{\boldsymbol{A}}_{jtk} \sim \text{Pois}(\delta_{jt}\vec{\boldsymbol{w}}_{jtk}), \overleftarrow{\boldsymbol{A}}_{jtk} \sim \text{Pois}(\delta_{jt}\overleftarrow{\boldsymbol{w}}_{jtk}) \tag{15}$$

This is the unique property of bi-conv-PGDS because of the bidirectional dynamic structure. We denote the convolutional representation of sentence $t$ in document $j$ as $\boldsymbol{W}_{jt} = \vec{\boldsymbol{W}}_{jt} + \overleftarrow{\boldsymbol{W}}_{jt} = (\boldsymbol{w}_{jt1}, \ldots, \boldsymbol{w}_{jtK}) \in \mathbb{R}_+^{K \times S_{jt}}$ Inspired by related development on inference for gamma Markov chains [43, 51, 60], we develop a backward-forward Gibbs sampling for forward feature representation $\vec{\boldsymbol{W}}_{jt}$, while a forward-backward Gibbs sampling for backward feature representation $\overleftarrow{\boldsymbol{W}}_{jt}$. Take $\vec{\boldsymbol{W}}_{jt}$ as an example, we should start with $\vec{\boldsymbol{W}}_{jT_j}$, because none of the other time-step depend on it in their priors. Via **P2**, we can marginalize over $\vec{\boldsymbol{W}}_{jT_j}$ and obtain the following equation

$$\vec{A}_{jT_jks_{jT_j}} \sim \text{NB}(\tau_0 \vec{\boldsymbol{\Pi}}_{k:} \sum_{s=1}^{S_{j,T_j}-1} \vec{\boldsymbol{w}}_{j,T_j-1,:s}/S_{j,T_j-1}, g(\vec{\zeta}_{jT_j})), \quad \vec{\zeta}_{jT_j} = \ln(1 + \frac{\delta_{jT_j}}{\tau_0}) \tag{16}$$

where $g(\zeta) = 1 - \exp(-\zeta)$. In order to marginalize over $\vec{\boldsymbol{W}}_{j,T_j-1}$, we introduce an auxiliary variable [13]

$$(\vec{x}_{jT_jks_{jT_j}}^{(2)} \mid -) \sim \text{CRT}(\vec{A}_{jT_jks_{jT_j}}, \tau_0 \vec{\boldsymbol{\Pi}}_{k:} \sum_{s=1}^{S_{j,T_j}-1} \vec{\boldsymbol{w}}_{j,T_j-1,:s}/S_{j,T_j-1}). \tag{17}$$

With **P3**, the joint distribution over $\vec{A}_{jT_jks_{jT_j}}$ and $\vec{x}_{jT_jks_{jT_j}}^{(2)}$ can be repressed as:

$$\begin{aligned} \vec{A}_{jT_jks_{jT_j}} &\sim \text{SumLog}(\vec{x}_{jT_jks_{jT_j}}^{(2)}, g(\vec{\zeta}_{jT_j})), \\ \vec{x}_{jT_jks_{jT_j}}^{(2)} &\sim \text{Pois}(\vec{\zeta}_{jT_j}\tau_0 \vec{\boldsymbol{\Pi}}_{k:} \sum_{s=1}^{S_{j,T_j}-1} \vec{\boldsymbol{w}}_{j,T_j-1,:s}/S_{j,T_j-1}). \end{aligned} \tag{18}$$

To marginalize over $\vec{\boldsymbol{W}}_{jT_j}$, again via **P1**, we augment $\vec{x}^{(2)}_{jT_jks_{jT_j}}$ as

$$\vec{x}^{(2)}_{jT_jks_{jT_j}} = \sum_{k_1=1}^{K} \vec{Z}_{jT_jkk_1s_{jT_j}}, \quad \vec{Z}_{jT_jkk_1s_{jT_j}} \sim \text{Pois}(\vec{\zeta}_{jT_j}\tau_0\vec{\Pi}_{kk_1}\sum_{s=1}^{S_{j,T_j-1}} \vec{w}_{j,T_j-1,k_1s}/S_{j,T_j-1}) \tag{19}$$

Since the transition weights $\sum_{k=1}^{K}\vec{\Pi}_{kk_1} = 1$, we have

$$\vec{Z}_{jT_j\cdot k_1s_{jT_j}} \sim \text{Pois}(\vec{\zeta}_{jT_j}\tau_0\sum_{s=1}^{S_{j,T_j-1}} \vec{w}_{j,T_j-1,k_1s}/S_{j,T_j-1}) \tag{20}$$

To get the likelihood for $\vec{w}_{j,T_j-1,k_1s_{j,T_j-1}}$, we further augment $\vec{Z}_{jT_j\cdot k_1s_{jT_j}}$ as

$$\vec{Z}_{jT_j\cdot k_1s_{jT_j}s_{j,T_j-1}} \sim \text{Pois}(\vec{\zeta}_{jT_j}\tau_0\vec{w}_{j,T_j-1,k_1s_{j,T_j-1}}/S_{j,T_j-1}) \tag{21}$$

Next, we summarize the information about the data at time-steps $T$ and $T-1$ as $\vec{u}_{j,T_j-1,ks_{jT_j}} = \vec{Z}_{jT_jk\cdot s_{j,T_j-1}} + \vec{A}_{j,T_j-1,ks_{j,T_j-1}}$, via **P1**, get the likelihood

$$\vec{u}_{j,T_j-1,ks_{j,T_j-1}} \sim \text{Pois}((\vec{\zeta}_{jT_j}\tau_0S_{jT_j}/S_{j,T_j-1} + \delta_{j,T_j-1})\vec{w}_{j,T_j-1,ks_{j,T_j-1}}) \tag{22}$$

With $\vec{u}_{j,T_j-1,ks_{j,T_j-1}}$, we can sample auxiliary $\vec{x}^{(2)}_{jT_j-1ks_{j,T_j-1}}$ via CRT distribution and re-express it in the same way with (16)(17)(18)

$$\vec{x}^{(2)}_{j,T_j-1,ks_{j,T_j-1}} = \text{Pois}(\vec{\zeta}_{j,T_j-1}\tau_0\Pi_{k:}\sum_{s=1}^{S_{j,T_j-2}} \vec{w}_{j,T_j-2,:s}/S_{j,T_j-2})$$

$$\vec{\zeta}_{j,T_j-1} = \ln(1 + \frac{\vec{\zeta}_{jT_j}\tau_0S_{jT_j}/S_{j,T_j-1} + \delta_{j,T_j-1}}{\tau_0}) = \ln(1 + S_{jT_j}\frac{\vec{\zeta}_{jT_j}}{S_{j,T_j-1}} + \frac{\delta_{j,T_j-1}}{\tau_0}) \tag{23}$$

Repeating the process all the way back to $t = 1$, it is able to marginalize over all latent feature representations $\{\vec{w}_{jtk}\}_{j=1,t=1,k=1}^{N,T,K}$.

**Sample forward feature representation $\vec{w}_{jtk}$:**  After sampling the auxiliary variable above, we sample $\vec{w}_{jtk}$ forward from $t = 1$ to $T$ using the gamma-Poisson conjugacy

$$(\vec{w}_{j1k}|-) \sim \text{Gam}(\vec{A}_{j1k} + \vec{Z}_{j2k\cdot:} + \tau_0\vec{v}_0, 1/(\delta_{j1} + \vec{\zeta}_{j2}\tau_0 + \tau_0))$$
$$(\vec{w}_{jtk}|-) \sim$$
$$\text{Gam}(\vec{A}_{jtk} + \vec{Z}_{j,t+1,k\cdot:} + \tau_0\vec{\Pi}_{k:}\sum_{s=1}^{S_{j,t-1}} \vec{w}_{j,t-1,:s}/S_{j,t-1}, 1/(\delta_{jt} + \vec{\zeta}_{j,t+1}\tau_0 + \tau_0)) \tag{24}$$

where $\vec{Z}_{j,T_j+1,k} = 0$, $\vec{\zeta}_{j,T_j+1} = 0$, $\vec{\zeta}_{jt} = \ln(1 + \frac{\delta_{jt}+\vec{\zeta}_{j,t+1}\tau_0}{\tau_0})$.

**Sample forward transition matrix $\vec{\Pi}$:**  Given (19) for each sentence and since $\sum_{k_1=1}^{K}\vec{\pi}_{k_1k} = 1$, we marginalize over $\vec{w}_{jt}$ and get

$$(\vec{Z}_{jt1k\cdot}, ..., \vec{Z}_{jtKk\cdot}|-) \sim \text{Multi}(\vec{Z}_{jt\cdot k\cdot}, (\vec{\pi}_{1k}, ..., \vec{\pi}_{Kk})). \tag{25}$$

With the Dirichlet-multinomial conjugacy, we have

$$(\vec{\pi}_k|-) \sim \text{Dir}(\vec{v}_1\vec{v}_k + \vec{Z}_{..1k\cdot}, ..., \vec{v}_K\vec{v}_k + \vec{Z}_{..Kk\cdot}) \tag{26}$$

For $\delta_{jt}, \vec{\beta}, \{\vec{v}_k\}_{k=1}^{K}$ and $\vec{\xi}$, we sample them in the same way with [43].

In parallel with backward-forward sampling for $\vec{\boldsymbol{W}}_{jt}$, we can sample $\overleftarrow{\boldsymbol{W}}_{jt}$ in the same way with $\vec{\boldsymbol{W}}_{jt}$, but towards the opposite direction.

# B  Scalable inference via stochastic gradient MCMC

While the computation of the Gibbs sampler can be accelerated inside each iteration, it requires processing all documents in each iteration and hence has limited scalability. For scalable inference, we update our global parameters by generalizing TLASGR-MCMC [17, 18], a stochastic gradient MCMC (SG-MCMC) algorithm that is proposed to sample simplex-constrained global parameters in a mini-batch learning setting and improve its sampling efficiency by preconditioning its gradient via the Fisher information matrix (FIM). More Specifically, after sampling auxiliary latent counts using parallel augmentable techniques as in Appendix A, the update of $\boldsymbol{D}$ with TLASGR-MCMC can be described as

$$
\begin{aligned}
(\boldsymbol{D}_k(:))_{i+1} = \Bigg\{ (\boldsymbol{D}_k(:))_i + \frac{\varepsilon_i}{M_k} \Big[ \Big( \rho \big( \boldsymbol{d}'_{\cdot k1}, \dots, \boldsymbol{d}'_{\cdot kV \cdot} \big)' + \eta \Big) \\
- (\rho d_{\cdot k \cdots} + \eta V F) \, (\boldsymbol{D}_k(:))_1 \Big] + N\left( 0, \frac{2\varepsilon_i}{M_k} \mathrm{diag}\left( (\boldsymbol{D}_k(:))_i \right) \right) \Bigg\}_{\angle}
\end{aligned}
\tag{27}
$$

where $[\cdot]_{\angle}$ denotes the simplex constraint and $M_k$ is calculated using the estimated FIM. $\boldsymbol{d}'_{\cdot kv}$ and $d_{\cdot k \cdots}$ come from (10) and (11). In addition, $\vec{\boldsymbol{\pi}}_k$, the $k$th column of the forward transition matrix $\vec{\boldsymbol{\Pi}}$, can be efficiently sampled as

$$
\begin{aligned}
(\vec{\boldsymbol{\pi}}_k)_{i+1} = \Bigg[ (\vec{\boldsymbol{\pi}}_k)_i + \frac{\varepsilon_i}{\vec{M}_k} \left[ (\rho \vec{z}_{:k\cdot} + \vec{v}) - (\rho \vec{z}_{\cdot k\cdot} + \vec{v}\cdot) (\vec{\boldsymbol{\pi}}_k)_i \right] \\
+ \mathcal{N}\left( 0, \frac{2\varepsilon_i}{\vec{M}_k} \left[ \mathrm{diag}(\vec{\boldsymbol{\pi}}_k)_i - (\vec{\boldsymbol{\pi}}_k)_i (\vec{\boldsymbol{\pi}}_k)_i^T \right] \right) \Bigg]_{\angle}
\end{aligned}
\tag{28}
$$

where $\vec{z}_{:k\cdot}$ and $\vec{z}_{\cdot k\cdot}$ come from the augmented latent counts $\vec{\boldsymbol{Z}}$ in (25) and $\vec{v}\cdot$ denotes the prior of $\vec{\boldsymbol{\pi}}_k$. The backward transition matrix $\overleftarrow{\boldsymbol{\Pi}}$ can be parallelly sampled in the same way. More details of TLASGR-MCMC can be found in Cong et al. [17] and Guo et al. [51]. The developed SG-MCMC algorithm for bi-conv-PGDS is described in Algorithm. 1.

---

**Algorithm 1** stochastic-gradient MCMC for bi-conv-PGDS

Input: Data mini-batches; Output: Global parameters of bi-conv-PGDS.
**for** $iter = 1, 2, \dots$ **do**
  \\* *Collect local information*
  Parallel bidirectional Gibbs sampling on the $i$th mini-batch for $\{\boldsymbol{d}_{jtv}\}_{j,t,v}$, $\{\vec{\boldsymbol{Z}}_{jtk_1k}\}_{j,t,k_1,k}$,
  $\{\overleftarrow{\boldsymbol{Z}}_{jtk_1k}\}_{j,t,k_1,k}$, $\{\vec{u}_{jtk}\}_{j,t,k}$, $\{\overleftarrow{u}_{jtk}\}_{j,t,k}$ with (10) (11) (25) (22);
  calculating for the $\left\{ \vec{\zeta}_{jt}, \overleftarrow{\zeta}_{jt} \right\}_{j,t}$ with (23);
  Parallel bidirectional Gibbs sampling for the $\left\{ \vec{\boldsymbol{W}}_{jt}, \overleftarrow{\boldsymbol{W}}_{jt} \right\}_{j,t}$ with (24);
  \\* *Update global parameters*
  **for** $k = 1, 2, \dots K$ **do**
    Update $M_k$ according to [20]; then $\{\boldsymbol{D}_k\}_k$ with (27); Update $\vec{M}_k$ and $\overleftarrow{M}_k$ according to [20];
    then $\{\vec{\boldsymbol{\pi}}_k\}_k$ and $\{\overleftarrow{\boldsymbol{\pi}}_k\}_k$ with (28);
  **end for**
  Update $\vec{\xi}, \overleftarrow{\xi}, \{\vec{v}_k, \overleftarrow{v}_k\}_k, \vec{\beta}, \overleftarrow{\beta}$ according to [43]
**end for**

---

# C  Hybrid SG-MCMC/auto-encoding variational inference

To allow for scalable inference and fast testing, we develop a hybrid SG-MCMC/autoencoding variational inference [47, 48] algorithm by combining TLASGR-MCMC [17, 18] and our proposed convolutional-recurrent variational inference network. In other words, in minibatch based each

iteration, we sample the global parameters $\boldsymbol{D}$, $\vec{\boldsymbol{\Pi}}$, and $\overleftarrow{\boldsymbol{\Pi}}$ with TLASGR-MCMC, while the parameters of the variational inference network, denoted by $\boldsymbol{\Omega}$, are updated via stochastic gradient descent (SGD).

More Specifically, after sampling auxiliary latent counts using parallel augmentable techniques, we update the global parameters $\boldsymbol{D}$, $\vec{\boldsymbol{\Pi}}$, and $\overleftarrow{\boldsymbol{\Pi}}$ with TLASGR-MCMC as in Appendix B. Given the global parameters, the task here is to optimize the parameters of the convolutional-recurrent variational inference network. As the usual strategy of variational inference, we achieve this optimization via SGD by minimizing the negative evidence lower bound (ELBO), which can be expressed as

$$
\begin{aligned}
L = \sum_{j=1}^{M} \sum_{t=1}^{T} \Big[ &\mathbb{E}_{q(\vec{\boldsymbol{W}}_{jt}, \overleftarrow{\boldsymbol{W}}_{jt})} \Big[ \ln p(\boldsymbol{x}_{jt} | \boldsymbol{D}, \vec{\boldsymbol{W}}_{jt}, \overleftarrow{\boldsymbol{W}}_{jt}) \Big] \\
&- \text{KL} \Big( q(\vec{\boldsymbol{W}}_{jt}, \overleftarrow{\boldsymbol{W}}_{jt}) || p(\vec{\boldsymbol{W}}_{jt} | \vec{\boldsymbol{\Pi}}, \vec{\boldsymbol{W}}_{j,t-1}) p(\overleftarrow{\boldsymbol{W}}_{jt} | \overleftarrow{\boldsymbol{\Pi}}, \overleftarrow{\boldsymbol{W}}_{j,t+1}) \Big) \Big]
\end{aligned}
\tag{29}
$$

$\text{KL}(q(\cdot)||p(\cdot)) = \mathbb{E}_{q(\cdot)} \log(q(\cdot)/p(\cdot))$ denotes the Kullback–Leibler (KL) Divergence from distribution $p(\cdot)$ to $q(\cdot)$. The proposed hybrid stochastic-gradient MCMC and autoencoding variational inference algorithm is described in Algorithm 2, which is implemented in PyTorch [62], combined with pyCUDA [63] for more efficient computation.

---

**Algorithm 2** Hybrid stochastic-gradient MCMC and autoencoding variational inference for bi-conv-PGDS

Set mini-batch size as $M$, the number of convolutional filters $K$ and hyperparameters;

Initialize inference model parameters $\boldsymbol{\Omega}$ and generative model parameters $\{\boldsymbol{D}_k\}_{k=1}^{K}$, $\vec{\boldsymbol{\Pi}}$ ,$\overleftarrow{\boldsymbol{\Pi}}$;

**for** $iter = 1, 2, ...$ **do**

    Randomly select a mini-batch of $M$ documents consisting of $T$ sentences to form a subset $\{\boldsymbol{x}_{i,1:T}\}_{i=1}^{M}$;

    Draw random noise $\{\epsilon_{i,t}\}_{i=1,t=1}^{M,T}$ from uniform distribution for sampling latent states;

    Calculate $\nabla_{\boldsymbol{\Omega}} L \left( \boldsymbol{\Omega}, \boldsymbol{D}, \vec{\boldsymbol{\Pi}}, \overleftarrow{\boldsymbol{\Pi}}; X, \epsilon_{i,t} \right)$ according to (29), and update $\boldsymbol{\Omega}$;

    Sample $\left\{ \vec{\boldsymbol{W}}_{jt}, \overleftarrow{\boldsymbol{W}}_{jt} \right\}_{j=1,t=1}^{M,T}$ from equation (4) in our paper via $\boldsymbol{\Omega}$, update $\boldsymbol{D}$ according to [20],

    update $\vec{\boldsymbol{\Pi}}$ and $\overleftarrow{\boldsymbol{\Pi}}$ according to (28) parallelly.

**end for**

---

## D   Datasets

Five large scale document datasets are used in this paper, including Reuters, ELEC, IMDB-2, IMDB-10 and yelp 2013. ELEC dataset [64] consists of electronic product reviews, which is part of a large Amazon review dataset. IMDB dataset is a benchmark dataset for sentiment analysis, where the task is to determine whether a movie review is positive or negative. Yelp is a restaurant review dataset. Reuters is a English news dataset and a multilabel-dataset. We use the standard ModApte splits [65] to transform Reuters to a single label dataset. Following [20, 57], for ELEC and IMDB-2, we use $50\%$ of the data for training and the remaining $50\%$ for testing. For other datasets, we use $80\%$ for training and the remaining $20\%$ for testing. The statistics of these five datasets are summarized in Table 5.

## E   Semi-supervised classification

In many practical scenarios, unlabeled data are abundant, however, there are not many practical cases where the potential of such unlabeled data is fully realized. Base on this condition, it is imperative to seek to complement scarcer but more valuable labeled data, to improve the generalization ability of supervised models. By ingesting unlabeled data, the model can learn to abstract latent representations that capture the semantic meaning of all available sentences irrespective of whether or not they are labeled. This can be done prior to the supervised model training, as a two-step process. Recently, many methods exploiting this idea have been wildly utilized and have achieved state-of-art performance in many tasks [55, 66–68].

Table 5: Data statistics: #s denotes the number of sentences (average and maximum per document), #w denotes the number of words (average and maximum per document), $V_{pre}$ denotes the number of words in the set of pre-trained word vectors.

| Data set | classes | documents | average#s | max#s | average#w | max#w | vocabulary | $V_{pre}$ |
|---|---|---|---|---|---|---|---|---|
| Reuters | 90 | 10789 | 6.6 | 248 | 144.3 | 2244 | 34397 | 30000 |
| ELEC | 2 | 50000 | 2.8 | 201 | 124 | 6000 | 52248 | 30000 |
| IMDB-2 | 2 | 50000 | 13.2 | 148 | 265 | 2800 | 95212 | 30000 |
| IMDB-10 | 10 | 135669 | 14.0 | 148 | 325.6 | 2802 | 115831 | 30000 |
| Yelp 2014 | 5 | 1125457 | 9.2 | 151 | 159.9 | 1199 | 476191 | 30000 |

**$0^{th}$ Topic**
| also 0.30 | like 0.62 | computer 0.74 |
| seems 0.20 | love 0.33 | all 0.05 |
| look 0.14 | battery 2e-3 | listening 0.02 |

**$1^{th}$ Topic**
| for 0.96 | my 0.94 | good 0.62 |
| people 4e-3 | some 0.02 | pocket 0.13 |
| in 4e-3 | appears 6e-4 | having 2e-3 |

**$2^{th}$ Topic**
| i 0.99 | have 0.98 | to 0.99 |
| they 8e-5 | haven't 0.012 | done 8e-5 |
| are 6e-5 | through 2e-4 | make 6e-5 |

**$3^{th}$ Topic**
| i 0.99 | am 0.98 | a 0.98 |
| more 2e-4 | own 7e-3 | given 3e-4 |
| son 1e-4 | supplied 1e-3 | fine 2e-4 |

**$4^{th}$ Topic**
| i 0.98 | would 0.98 | recommend 0.93 |
| might 8e-4 | then 0.02 | suggest 0.06 |
| worth 3e-4 | highly 2e-4 | edge 4e-3 |

**$5^{th}$ Topic**
| i 0.99 | thought 0.39 | it 0.99 |
| look 2e-4 | used 0.22 | notice 1e-4 |
| falling 1e-4 | ordered 0.12 | well 9e-5 |

**$6^{th}$ Topic**
| happy 0.43 | with 0.98 | the 0.98 |
| pleased 0.35 | roll 2e-4 | recorded 2e-4 |
| well 0.16 | recharge 1e-4 | said 2e-4 |

**$7^{th}$ Topic**
| it 0.99 | 's 0.97 | a 0.98 |
| so 1e-4 | into 0.01 | attachment 1e-4 |
| rarely 1e-4 | through 4e-4 | packaged 1e-4 |

**$8^{th}$ Topic**
| all 0.74 | are 0.31 | fine 0.70 |
| they 0.21 | dvd 0.29 | trying 0.23 |
| unit 1e-3 | screen 0.20 | ear 6e-3 |

**$9^{th}$ Topic**
| i 0.99 | was 0.99 | very 0.71 |
| deliver 1e-4 | found 1e-4 | told 0.09 |
| risk 6e-5 | opened 9e-5 | hoping 0.07 |

**$10^{th}$ Topic**
| i 0.99 | did 0.62 | not 0.99 |
| more 4e-4 | could 0.16 | plug 2e-4 |
| might 3e-4 | went 0.14 | broadcast 9e-5 |

**$11^{th}$ Topic**
| i 0.93 | bought 0.91 | this 0.85 |
| less 0.03 | than 0.06 | an 0.12 |
| more 0.02 | eject 6e-5 | one 3e-4 |

**$12^{th}$ Topic**
| this 0.99 | is 0.99 | a 0.98 |
| tv 4e-4 | matter 3e-4 | haven't 2e-4 |
| laptop 2e-4 | type 3e-4 | recall 2e-4 |

**$13^{th}$ Topic**
| have 0.98 | found 0.54 | problem 0.96 |
| opened 7e-4 | any 0.31 | broken 3e-4 |
| lap 2e-4 | guess 3e-3 | none 3e-4 |

**$14^{th}$ Topic**
| you 0.98 | can 0.93 | back 0.60 |
| they 2e-3 | won't 0.04 | hear 0.21 |
| whole 2e-4 | would 0.03 | beat 0.07 |

Figure 7: Example phrase-level topics from ELEC learned by conv-PGDS.

As a complex pre-trained methods, BERT is pre-trained on large scale data and achieved state-of-the-art performance in many tasks. In the semi-supervised experiment of this paper, we first pre-train BERT with unlabeled data using masked ML (MLM) tasks, which is designed by masking some percentage of the input tokens at random, and then predict those masked tokens. Then, we train it on labeled data for document classification.

Our proposed convolutional-recurrent autoencoding framework can also be applied to semi-supervised learning from document. Instead of pre-training a fully unsupervised model as BERT, we simultaneously train a sequence autoencoder and a supervised model. In principle, by using the joint training strategy as in Section 3 in our paper, the learned document embedding vector will preserve both reconstruction and classification ability. Specifically, we consider the following objective

$$L^{semi} = \sum_{d \in \{\mathcal{D}_l + \mathcal{D}_u\}} L_g(d) + \xi \sum_{d \in \mathcal{D}_l} L_s(d) \qquad (30)$$

which is a modification of our supervised loss, where $\mathcal{D}_l$ and $\mathcal{D}_u$ denote the set of labeled and unlabeled data, respectively.

# F  Additional experimental results

We list the top 15 phrase-level topics from ELEC learned by conv-PGDS in Fig. 7, which are related to the corresponding transition matrix in Fig. 3 in our paper. It is clearly that the words in different convolutional filters can be combined into different phrase-level topics.

In addition to conv-PGDS, we also visualize the inferred convolutional filters and transition matrices of bi-conv-PGDS. Fig. 8 shows the top 20 topics learned by bi-conv-PGDS on ELEC, whose

**0th Topic**

| i | 0.99 | have | 0.62 | to | 0.99 |
|---|---|---|---|---|---|
| allow | 6e-5 | listen | 0.33 | doesnt | 9e-5 |
| being | 6e-5 | targus | 2e-3 | pouch | 9e-5 |

**1th Topic**

| this | 0.96 | product | 0.94 | is | 0.92 |
|---|---|---|---|---|---|
| after | 4e-3 | volume | 0.02 | my | 3e-3 |
| no | 4e-3 | deal | 6e-4 | when | 2e-3 |

**2th Topic**

| if | 0.85 | you | 0.98 | are | 0.52 |
|---|---|---|---|---|---|
| allows | 0.06 | two | 0.012 | can | 0.44 |
| believe | 0.05 | bargin | 2e-4 | market | 6e-5 |

**3th Topic**

| i | 0.98 | 'm | 0.94 | the | 0.97 |
|---|---|---|---|---|---|
| prior | 0.01 | liked | 0.05 | kind | 0.01 |
| neighbor | 6e-5 | opened | 2e-4 | tried | 6e-3 |

**4th Topic**

| i | 0.99 | love | 0.38 | it | 0.98 |
|---|---|---|---|---|---|
| ink | 5e-4 | think | 0.26 | layout | 6e-4 |
| button | 1e-4 | saw | 0.20 | finally | 9e-5 |

**5th Topic**

| better | 0.98 | car | 0.78 | than | 0.98 |
|---|---|---|---|---|---|
| ago | 8e-4 | piece | 0.12 | still | 6e-3 |
| live | 3e-4 | life | 0.04 | mail | 4e-3 |

**6th Topic**

| i | 0.96 | thought | 0.69 | it | 0.95 |
|---|---|---|---|---|---|
| child | 0.02 | must | 0.22 | say | 0.01 |
| charm | 1e-4 | allow | 2e-3 | install | 9e-3 |

**7th Topic**

| one | 0.39 | of | 0.89 | the | 0.99 |
|---|---|---|---|---|---|
| most | 0.30 | desk | 0.02 | dozen | 1e-4 |
| any | 0.18 | unplug | 2e-3 | annoying | 9e-5 |

**8th Topic**

| can | 0.63 | recommend | 0.98 | this | 0.98 |
|---|---|---|---|---|---|
| highly | 0.15 | week | 2e-4 | used | 2e-4 |
| not | 6e-3 | two | 1e-4 | medium | 2e-4 |

**9th Topic**

| but | 0.99 | it | 0.97 | was | 0.98 |
|---|---|---|---|---|---|
| picture | 1e-4 | big | 0.01 | last | 1e-4 |
| refund | 1e-4 | plenty | 4e-4 | i | 1e-4 |

**10th Topic**

| really | 0.94 | like | 0.99 | that | 0.97 |
|---|---|---|---|---|---|
| wanted | 8e-3 | dvd | 9e-4 | pleased | 3e-3 |
| decided | 1e-3 | front | 2e-4 | ever | 6e-3 |

**11th Topic**

| i | 0.99 | would | 0.32 | not | 0.97 |
|---|---|---|---|---|---|
| concerned | 2e-4 | did | 0.28 | before | 3e-3 |
| table | 1e-4 | should | 0.20 | strong | 6e-4 |

**12th Topic**

| it | 0.97 | is | 0.96 | a | 0.91 |
|---|---|---|---|---|---|
| window | 0.01 | adjust | 0.02 | difficult | 0.05 |
| buying | 6e-5 | built | 9e-5 | so | 7e-4 |

**13th Topic**

| the | 0.99 | price | 0.82 | is | 0.99 |
|---|---|---|---|---|---|
| picture | 4e-4 | usb | 0.16 | remote | 2e-4 |
| wrist | 3e-4 | button | 4e-3 | port | 9e-5 |

**14th Topic**

| for | 0.98 | the | 0.91 | price | 0.85 |
|---|---|---|---|---|---|
| micro | 3e-4 | several | 0.06 | year | 0.12 |
| mind | 2e-4 | dony | 3e-3 | about | 3e-4 |

**15th Topic**

| only | 0.97 | happy | 0.92 | with | 0.97 |
|---|---|---|---|---|---|
| other | 3e-3 | come | 0.05 | believe | 2e-3 |
| very | 2e-3 | came | 6e-5 | charm | 3e-4 |

**16th Topic**

| just | 0.99 | first | 0.90 | version | 0.78 |
|---|---|---|---|---|---|
| long | 4e-4 | memory | 0.03 | card | 0.12 |
| close | 2e-4 | perfectly | 0.02 | day | 0.04 |

**17th Topic**

| i | 0.98 | am | 0.54 | this | 0.96 |
|---|---|---|---|---|---|
| without | 7e-4 | purchased | 0.31 | two | 3e-4 |
| road | 2e-4 | gauss | 3e-3 | support | 3e-4 |

**18th Topic**

| a | 0.98 | lot | 0.93 | of | 0.97 |
|---|---|---|---|---|---|
| ended | 2e-3 | pair | 0.04 | advantage | 2e-3 |
| casing | 2e-4 | couple | 3e-3 | focus | 7e-4 |

**19th Topic**

| was | 0.98 | a | 0.98 | replacement | 0.86 |
|---|---|---|---|---|---|
| unless | 0.01 | memory | 4e-4 | told | 0.11 |
| over | 2e-4 | operate | 3e-4 | yet | 7e-4 |

Figure 8: Example phrase-level topics from ELEC learned by bi-conv-PGDS.

corresponding forward and backward matrices are shown in Fig. 9. Moreover, we visualize some examples of forward and backward transition relations between different topics in Fig. 10 and Fig. 11 respectively. From the examples shown in Fig. 10, we can find that forward transition relations learned by bi-conv-PGDS are similar with these learned by conv-PGDS. It is particularly interesting to notice that the backward transition relations are partly reverse to the forward ones. Taking Fig. 10 (a) as a forward transition example, filter 17 on "i purchased this" is the phrase focusing on how to get the product and it is more likely to be transited to filter 1, 12, 13 that are mainly about "product description". As for the backward transition, in Fig. 10 (a), filter 1 on "this product is" is the phrase on "product description", it is more likely to to activate transitions to filter 6, 8, 12 that are mainly about "how to get the product" or "product description". In addition, there are also some filters that other filters are more likely to active a transition to themselves, not matter forward and backward, such as filter 12 shown in both Fig. 10 (b) and Fig. 11 (b).

Figure 9: Forward transition matrix $\overrightarrow{\Pi}$ (a) and backward transition matrix $\overleftarrow{\Pi}$ (b) for top 20 topics from ELEC learned by bi-conv-PGDS.

Following the same way of Fig. 4 in our main manuscript, we provide more visualized examples of the hierarchical attention learned by our model and compare them with CPFA equipped with attention in Figs. 12 and 13.

As introduced before, the reason why we develop a parallelized Gibbs sampler as well as a hybrid SG-MCMC/VI is that both of them have their own advantages. We compare the accuracy of text classification and the corresponding testing time in Table 6. As we can see, the use of encoder makes our model fast in testing time, but leads to a tradeoff in accuracy.

**Figure 10 (a)**

**17th Topic** — i 0.98, am 0.54, this 0.96 | without 7e-4, purchased 0.31, two 3e-4 | road 2e-4, gauss 3e-3, support 3e-4

**1st Topic** — this 0.96, product 0.94, is 0.92 | after 4e-3, volume 0.02, my 3e-3 | no 4e-3, deal 6e-4, when 2e-3

**12th Topic** — it 0.97, is 0.96, a 0.91 | window 0.01, adjust 0.02, difficult 0.05 | buying 6e-3, built 9e-5, so 7e-4

**13th Topic** — the 0.99, price 0.82, is 0.99 | picture 4e-4, usb 0.16, remote 2e-4 | wrist 3e-4, button 4e-3, port 9e-5

**Figure 10 (b)**

**4th Topic** — i 0.99, love 0.38, it 0.98 | ink 5e-4, think 0.26, layout 6e-4 | button 1e-4, saw 0.20, finally 9e-5

**12th Topic** — it 0.97, is 0.96, a 0.91 | window 0.01, adjust 0.02, difficult 0.05 | buying 6e-3, built 9e-5, so 7e-4

**3rd Topic** — i 0.98, 'm 0.94, the 0.97 | prior 0.01, liked 0.05, kind 0.01 | neighbor 6e-5, opened 2e-4, tried 6e-3

**17th Topic** — i 0.98, am 0.54, this 0.96 | without 7e-4, purchased 0.31, two 3e-4 | road 2e-4, gauss 3e-3, support 3e-4

Figure 10: Two example forward transition relations between topics from ELEC learned by bi-conv-PGDS (better understood together with Fig.9 (a)).

**Figure 11 (a)**

**1st Topic** — this 0.96, product 0.94, is 0.92 | after 4e-3, volume 0.02, my 3e-3 | no 4e-3, deal 6e-4, when 2e-3

**12th Topic** — it 0.97, is 0.96, a 0.91 | window 0.01, adjust 0.02, difficult 0.05 | buying 6e-3, built 9e-5, so 7e-4

**6th Topic** — i 0.96, thought 0.69, it 0.95 | child 0.02, must 0.22, say 0.01 | charm 1e-4, allow 2e-3, install 9e-3

**8th Topic** — can 0.63, recommend 0.98, this 0.98 | highly 0.15, week 2e-4, used 2e-4 | not 6e-3, two 1e-4, medium 2e-4

**Figure 11 (b)**

**4th Topic** — i 0.99, love 0.38, it 0.98 | ink 5e-4, think 0.26, layout 6e-4 | button 1e-4, saw 0.20, finally 9e-5

**12th Topic** — it 0.97, is 0.96, a 0.91 | window 0.01, adjust 0.02, difficult 0.05 | buying 6e-3, built 9e-5, so 7e-4

**9th Topic** — but 0.99, it 0.97, was 0.98 | picture 1e-4, big 0.01, last 1e-4 | refund 1e-4, plenty 4e-4, i 1e-4

**1st Topic** — this 0.96, product 0.94, is 0.92 | after 4e-3, volume 0.02, my 3e-3 | no 4e-3, deal 6e-4, when 2e-3

Figure 11: Two example backward transition relations between topics from ELEC learned by bi-conv-PGDS (better understood together with Fig.9 (b)).

**attn-CPFA** GT:4 Prediction: 5

Normally I love chipotle and I would rate it 5 stars .

The ingredients are scrumptious and the experience is sublime .

However , at this restaurant not all of the staff was on their game .

**attn-bi-conv-PGDS** GT:4 Prediction: 4

0.10 Normally I love chipotle and I would rate it 5 stars .

0.24 The ingredients are scrumptious and the experience is sublime .

0.66 However , at this restaurant not all of the staff was on their game .

**attn-CPFA** GT:5 Prediction: 3

What I hate about buffets in general is that the food is usually cooked poorly when cooked in large quantities

But that is totally not the case here .

All the meat and fish I sampled were cooked perfectly .

**attn-bi-conv-PGDS** GT:5 Prediction: 5

0.12 What I hate about buffets in general is that the food is usually cooked poorly when cooked in large quantities

0.46 But that is totally not the case here .

0.42 All the meat and fish I sampled were cooked perfectly .

Figure 12: Examples of attention for Yelp'14 learned by attn-CPFA and attn-bi-conv-PGDS.

**attn-CPFA** GT:0 Prediction: 1

I really enjoyed using these headphones .

They are comfortable , lightweight and they sound great .

However , they are not made well at all .

If you use these daily , the left ear will short out after about a month of use .

**attn-bi-conv-PGDS** GT:0 Prediction: 0

0.08 I really enjoyed using these headphones .

0.13 They are comfortable , lightweight and they sound great .

0.44 However , they are not made well at all .

0.35 If you use these daily , the left ear will short out after about a month of use .

Figure 13: Examples of attention for ELEC learned by attn-CPFA and attn-bi-conv-PGDS.

Table 6: Comparison of the testing times (seconds) with batch-size 25.

| Methods | Accuracy | | | | Testing time in seconds | | | |
|---|---|---|---|---|---|---|---|---|
| | Reuters | ELEC | IMDB-2 | IMDB-10 | Reuters | ELEC | IMDB-2 | IMDB-10 |
| bi-conv-PGDS (TLASGR-MCMC) | $78.0 \pm 0.7$ | $84.5 \pm 0.8$ | $84.0 \pm 0.8$ | $37.9 \pm 0.4$ | 17.35 | 15.07 | 25.61 | 27.52 |
| bi-conv-PGDS (hybrid SG-MCMC and VI) | $76.8 \pm 1.0$ | $83.7 \pm 1.2$ | $82.6 \pm 1.1$ | $36.9 \pm 0.7$ | 0.15 | 0.14 | 0.17 | 0.20 |