[Reviews · NeurIPS 2020]

Review 1

Summary and Contributions: The paper develops a fully probabilistic model for natural text, matching roughly the state-of-the-art transformer models with more interpretable model that uses fully Bayesian inference.

Strengths: The paper is impressive. It manages to avoid most of the simplifications the probabilistic modelling community has accustomed to using, by simply not being afraid to address the challenge of adding more structure for the model. While the key element of convolutional components is not new, it is still very recent and is here used to build a serious model of complex text collections. The inference algorithms also build on recent techniques, and the attention capability is a nice bonus. The empirical experiments are comprehensive and demonstrate very good performance against the competition.

Weaknesses: I do not see any notable weaknesses.

Correctness: The method seems correct, and the empirical evaluations are comprehensive and properly carried out. The claims regarding performance compared to transformers are reasonable and supported by the evidence; I do not think the specific relative order of the proposed method and DocBERT is even relevant -- the real value of this work is in demonstrating how far fully Bayesian modelling of text can be brought when done properly, and by doing this the paper provides solid basis for followup work in the field.

Clarity: The paper is well written and easy to read, and includes well designed and useful illustrations (Fig 1, 2 and 4), with the only minor issue being their small scale dictated by the page limit.

Relation to Prior Work: Both earlier probabilistic models and the parallel research in deep learning for natural language are well covered, and the specific contribution is clear.

Reproducibility: Yes

Additional Feedback: Additional comments after rebuttal: Thank you for a clear rebuttal. I have no need to change my positive evaluation of the work. The broader impact statement of this paper was more thoughtful and comprehensive than in any of the other papers I reviewed, despite the contribution being highly technical. Kudos for that.


Review 2

Summary and Contributions: The paper studies the problem of document representation, the fundamental problem in text analysis and language modeling. Specifically, it follows the line of Poisson factor analysis, i.e., the probabilistic topic model. To incorporate the document-sentence-word hierarchical structure, it proposes the convolutional Poisson gamma dynamical system, which employs word-level convolution and sentence-level transition to model the phrase-level topics and topical evolution along with sentences. Besides, for efficient model training and inference, it proposes a hybrid inference algorithm based on the SG-MCMC and a convolutional-recurrent variational inference network. Finally, extensive experiment s are conducted to verify the effectiveness of the proposed methods in unsupervised and supervised settings. The main contributions lie in the (bi-)conv-PGDS and the corresponding scalable training algorithm.

Strengths: 1. The studied topic is a key research problem in text analysis, which is related to the NeurIPS community. 2. The authors are aware of the evolution of the related areas, and the proposed method could be seen as the marriage of probabilistic topic models and deep neural networks, which is of great novelty. 3. The proposed methods are clearly introduced with comprehensive derivation (more details are in the supplementary files) and comparison with previous models. 4. Extensive experiments are conducted on various text classification benchmarks, and the results are compared with typical and SOTA baselines.

Weaknesses: 1. The motivation is conceptually described, and an example could help reader understand how the hierarchical structure benefits the document representation. 2. A standalone literature review part could be better. 3. The model description could be improved, e.g., the generative process is in detail but presenting such process in separate steps should be better for understanding, too many symbols and a notation table could be better. 4. The evaluation task only contains text classification and more tasks should be included. Besides, the paper does not provide enough details to reproduce the results (the demo code is not enough, some suggestions/guidance about the model setting for different types of documents could be more helpful).

Correctness: The claims and the proposed method are both clearly demonstrated, and I do not find any obvious problems in the model description and evaluation.

Clarity: The paper is well-organized, clearly written, and easy to understand.

Relation to Prior Work: There is no standalone related work part, but the paper discusses how the current research relates to previous work in the following aspects: 1. It reviews the related literature in the introduction. 2. It discusses the relationship between the proposed conv-PGDS with CPFA and PGDS at the end of Section 2.1. 3. Similarly, the training and inference algorithms are based on existing TLASGR-MCMC. 4. The experiments also include the above-mentioned SOTA methods as baselines.

Reproducibility: No

Additional Feedback: I've read the author response, and the recommendation score remains.


Review 3

Summary and Contributions: The paper combines convolution operators (Wang et al. ICML19), Poisson gamma dynamical models (Guo et al. NIPS 18), main techniques rising from (Zhou et al. JLMR 16) to formulate a hierarchical Bayesian model and its bidirectional extension, to capture both word- and sentence-structure. They demonstrate the proposed methods in unsupervised and supervised learning. Compared with DocNADE, the hierarchical Bayes models and the related deep extensions included here, their model seems to achieve higher accuracy.

Strengths: The idea of reflecting these probabilistic methods all together in a novel hierarchical Bayes model, looks interesting. Specifically, the bidirectional extension of the convolutional PGDS is novel. The proposed models were shown to achieve higher accuracy although not that much.

Weaknesses: Although all these probabilistic techniques are first reflected all together, Im mainly concerning the novelty of doing so. The comparisons to unsupervised models seems to be weak as only related hierarchical models were included. It is not sufficient to leave out deep neural nets and only consider hierarchical models, to reach broad audience in a venue like neurips. Why the accuracy of those baseline models lack standard deviations? It is not clear how to set or to sample those hyper-parameters like \epsilon, \gamma, \tau. It lack sensitive analysis of hyperparameters in main context and supplement.

Correctness: -The hierarchical model looks reasonable as they are motivated. The derivation of MCMC sampling scheme and its stochastic extension are solid.

Clarity: The presentation of the model is clear. It is better to discuss how the paper relate to a broad class of related methods (deepNNs) in one session.

Relation to Prior Work: They have discussed how their model relate to, and the limitations of previous model. It motivates them to combine all these techniques in a new hierarchical model. I am wondering why the deep neural nets are not chose for comparison in the experiments. The claims \L40-54 seem to be weak by explaining deep NNs cannot explain the semantic meanings of the learned parameters.

Reproducibility: Yes

Additional Feedback: Update after the reply: The authors have clarified some of the points I raised. Nonetheless, their reply does not change my assessment of the novelty. Thus I will keep my score. In addition, I think it is still necessary to revise the final version according to the detailed comments of all the reviews.


Review 4

Summary and Contributions: The paper presents a new hierarchical Bayesian model -- convolutional Poisson-Gamma Dynamical Systems (conv-PGDS) -- for generating the observed words in a document corpus. The model explicitly captures the natural document-sentence-word structure of such datasets via a carefully-chosen latent variable model. Globally, the model assumes there are K "topic filters", D_1, ... D_K, which are distributions over 3-grams from a finite size vocabulary (size V). Each "topic" (indexed by k) has an appearance probability weight v_k > 0 for appearing in a document, and we define transition probability vectors \pi_k Given this global structure, the model generates each document iid. To generate a document j, we use a Gamma dynamical system (with transitions \pi) to obtain a sequence of un-normalized membership "weight embeddings", w_j1 ... w_jT, one for each sentence (indexed by t). Each weight embedding vector w_jt indicates the relative weight of topic k across all words in the sentence t. To generate the t-th sentence of document j, we convolve that sentence's weights for topic k -- w_jtk -- with the topic filter D_k as in Equation 1 to obtain the mean vector (up to a scale factor) for a Poisson r.v. M_jt, which is thresholded to obtain the one-hot representation of the observed sentence X. This modeling direction has roots in widely-used topic models such as latent Dirichlet allocation [5] and Poisson factor analysis [7], which assume a "bag of words" model. This paper overcomes the restrictive orderless assumption of "bag-of-words". Previously, the 3-grad convolution approach to word-level structure was used in the Conv-Poisson Factor Analysis model of Wang et al (ICML 2019). However, that paper did not address sentence-level structure, assuming each sentence was drawn iid given document-level information. The claimed contributions of this paper are: * A new hierarchical latent variable model for document-sentence-word datasets that captures important non-iidsentence-level structure (via a dynamical system) and non-iid word-level structure (via a 3-gram convolution). * An extension of this model to bidirectional data processing. * An extension of this model to supervised and semi-supervised tasks via an attention-based document-level feature * Empirical demonstrations for these innovations in improving supervised document classification on several benchmark datasets (e.g. ELEC product review classification, IMDB movie review sentiment classification) Parameter learning is done via a scalable Gibbs sampling approach (Sec 3.1), which is made more efficient with an approximate local variable posterior obtained via an amortized encoder architecture (Sec. 3.2; inspired by VAEs).

Strengths: * Extensive empirical comparisons to various methods for document classification, spanning several representations (bag-of-words, word-order methods, word-order and sentence-order methods) and methodologies (generative, discriminative RNNs/CNNs, transformers). * Demonstrations of quantitative improvements in semi-supervised learning (even with 5% labels) on IMDB-10 * Demonstrations of competitive classifier performance compared to BERT transformer models (only a few points different in supervised accuracy while using far fewer parameters) * Qualitative demonstration of the benefits of bidirectional-convolution in Figure 4 * Inclusion of model size and runtime-per-iteration as additional axes of comparison (useful for a well-rounded assessment)

Weaknesses: * The hybrid SG-MCMC + variational encoder approach to parameter learning and inference seems poorly justified: what guarantees can we claim about this hybrid's approximation quality?. I'm sure it does something useful at least in early iterations, but it's not clear why we should attempt the SG-MCMC for global parameters versus just do hill-climbing with a pure ELBO objective for all parameters and encoder weights * Reproducible details are needed (how to set gradient optimization hyperparameters, how to pick key hyperparameters for baseline methods, etc.) * The quality of the learned "topic filters" shown in Fig 2 seems suspect. Many of the topics shown have basically nonsense phrases as one of their most common paths (e.g. "lap guess none" in 13th topic, "rarely though packaged" in Topic 26, "plased roll recorded" in 6th topic). These are almost less coherent than many topics found via Latent Dirichlet Allocation. * Evaluation focuses entirely on supervised document classification. While this is a useful task, it would be interesting to also consider the quality of the learned generative model.... can it produce coherent sentences, or assign high probability to real sentences over orderless ones? * The contribution of the encoder is not quantified. If you already have a Gibbs sampler that can be parallelized across documents, how much more efficient is it to do the encoder?

Correctness: The presented "practical" algorithm is a hybrid of MCMC and VI. They do SG-MCMC on the global model parameters, but then use an encoder to do fast per-document inference (even though they have separately claimed to develop a parallelized Gibbs sampler). I understand this is done for scalability, but I'm concerned that in this hybrid setting there are likely few (if any) guarantees and little we can say about what kind of solution the model converges to, which makes this method hard to justify except that it is practically useful, and I'd like the text to acknowledge this. Specifically, we cannot be confident the asymptotic guarantees of MCMC would ever kick in if the local posteriors do not converge (can we?), and it is not clear that the amortized encoder will produce a high-quality approximation of the posterior even for a fixed set of model parameters without more info about how the hybrid is performed (e.g. is that optimization run to convergence? how loose is the ELBO bound?). That said, no MCMC method on real-sized data ever really "converges" anyway, so this isn't a dealbreaker, but worth acknowledging.

Clarity: Overall the ideas were clearly communicated. One minor suggestion would be to move Figure 2 and Figure 3 (which visualize learned model structure) closer together to help readers easily interpret both together.

Relation to Prior Work: The paper adequately cites prior work and places itself in context. I don't see major issues here, thought the text could be more clear how the Conv-PFA model from [18] discussed as a similar model to the one here is related to the CPGBN model from [18] compared in the experiments. Key difference seems to just be using multiple stochastic layers (not just one), but this should be clearly stated.

Reproducibility: No

Additional Feedback: Review Summary --------------------- Overall I like this work, as it presents a useful hierarchical Bayesian model that captures multi-level document-sentence-word structure in human-generated natural language corpora (avoiding previous iid assumptions about sentence-order and word-order). With appropriate supervised extensions, the proposed model reaches remarkable performance on document-level classification tasks (in both unsupervised + downstream linear classifier, fully-supervised, and semi-supervised settings) that is competitive with state-of-the-art. If rebuttal can address my concerns about the hybrid SG-MCMC/variational model fitting methodology, reproducibility, generative model evaluation, and parameter interpretability, I would be happy to accept as I think this would inspire productive discussions at NeurIPS for the latent variable modeling subcommunity and mark a fundamental step forward in using generative latent variable models for document classification tasks. Post Rebuttal Summary --------------------------- After reading rebuttal and other reviews, I continue to argue for acceptance. I appreciate the careful rebuttal offered by the authors. RE interpretability: Thanks for clarifying the paths I was looking at were low probability. This makes sense to me. Please update the figures to show that these are low-probability in the revised paper. RE updated results on sentence order, document clustering: Thanks for including! Glad to see this is sensible and that the paper will not just have supervised performance metrics anymore. RE hybrid inference: I totally understand the practical needs for speed in prediction time. I would like to see more discussion in the paper about why this hybrid is a sensible strategy even when acknowledging it doesn't have guarantees. The speed / accuracy tradeoff in Table 2 is also important to include / highlight. Experimental Issues ------------------- ## E1: Reproducibility concerns Table 3 should probably include a model size column (like Table 1 does). Somewhere in the supplement, I would strongly suggest the authors add much more reproducible details about experiments. For each method: * what is the model size (number of topics, etc)? * what hyperparameter settings are used? how were these selected? * was a validation set used to tune/select hyperparameters? * what training settings were used (step sizes? gradient-descent method? convergence criteria?, etc.) * where is the code found? was this an internal reimplementation or use of a published package or a number from some existing paper? It does seem several raw numbers are more-or-less copied from [18]. But there should still be clear pointers from this work to the relevant information in [18] if it exists there. ## E2: Experiments do not assess the generative model enough There is little qualitative or quantitative evaluation of the proposed model as a *generative* model. I would have liked to see a bit more on this front, either via samples of text or via heldout likelihoods or something. The model is appealing because it captures between-sentence and within-sentence structure.... however only the attention visualization in Figure 4 starts to get at evaluating how well the methods work, and uses what could be "cherry picked" examples rather than rigorous evaluation. Looking at Table 1, we see that the unsupervised CPGBN [18] (which does NOT model sentence-to-sentence structure) gets *almost* as good classification (basically within the error bars of the single-directional conv-PGDS) on 3 out of 4 datasets. I wonder if more evidence can be provided that such innovations are really worth it. ## E3: Experiments do not assess the proposed encoder enough If there is already a fast parallelized Gibbs sampler for local document structure, how much better is the encoder (in runtime)? What tradeoff is there in accuracy for this scalability? ## E4: Uncertainty about performance metrics needs clarification Some methods in Table 1 and Table 2 have uncertainty communicated via a +/- number. But I cannot find a description of what this is quantifying (either in caption or in main text). Is this variability over random initializations of the method? Variability over randomly selected test sets? Something else? Is this reporting one standard error, or something else? Line-by-line feedback --------------------- Eq. 1: Need to denote clearly that w_j1k and w_jtk are *vectors* over the S_jk words. Eq. 1: I think the w generation notation could be simplified. there are inconsistent commas and colons in the subscripts of w (this notation is not defined), it is not clear if \pi is a product or a variable at first glance, etc. Eq. 1: Should also clarify which version of Gamma parameterization you are using: does Gamma(a,b) have mean of a/b or mean of a*b? Eq. 1: Should also clarify why write in terms of an integer r.v. M that just gets thresholded? Why not define a Bernoulli distribution directly on the observed X? (I guess there's an auxiliary variable trick used later?) Line 166: Motivate why the Weibull is a good choice. Reparameterization trick? Line 169: should say "... are the parameters of $q(w)$ ..." (missing q) Line 198: Didn't we already use \xi notation to indicate self-transition earlier in Eq 2?

[Author Response · NeurIPS 2020]

We thank the reviewers for their valuable comments and suggestions. We first respond to all: 1) We set the same network structure for all models in supervised experiments: 200, 200-100, and 200-100-50 for one-, two-, and three-layer models, respectively. 2) We fix the hyperparameters of our models as $\tau_0 = 1, \epsilon_0 = 0.1, \gamma_0 = 0.1, \eta = 0.05$ for all experiments; the performance is not sensitive to these hyperparameters, an usual advantage of hierarchical Bayesian models.

| Methods | ELEC | |
|---|---|---|
| | Accuracy | NMI |
| PGBN | 71.4 | 60.8 |
| CPGBN | 77.8 | 65.1 |
| CPGDS | 78.6 | 66.2 |

Table 1: Clustering performance comparison.

**To R1:** Thank you for your positive feedback, which really encourages us to continue our efforts along this promising direction! **To R2 & R4:** 1) To verify the efficiency of our generative model, additional tasks on document clustering and sentence and document likelihood evaluations have been included. Following Cai et al. (TPAMI 2011), we use accuracy and NMI to evaluate document clustering performance, as shown in Table 1 (we only include dataset ELEC given space constraint; we will add more methods on more datasets), which further verifies the advantages of CPGDS. We estimate the likelihood of sentence with shuffled word order. Fig. 1 (left) shows the likelihood decreases as the shuffling rate increases, indicating CPGDS provides a higher confidence on real sentences than orderless ones. We further estimate the likelihood of document with shuffled sentence order and observe similar behaviors in Fig. 1 (right).

**To R2:** Example sentences in Fig. 4 are provided to illustrate the point that introducing the relationships between different sentences can help improve the accuracy of the sentiment-level judgments for the whole document, which confirms our motivations.

**To R3:** 1) Combining CFPA and PGDS into a coherent statistical model requires addressing several technical challenges, such as how to handle variable sentence lengths, avoid cutting off backward message passing, and speed up Gibbs sampling. 2) First, we have compared our

Figure 1: left: likelihood of shuffled sentence; right: likelihood of shuffled document

model with a wide variety of topic models and unsupervised generative models in unsupervised experiments. To the best of our knowledge, except for DocNADE that is already included for comparison, there are few deep NN based probabilistic models for unsupervised document modeling. Second, in supervised experiments, we have compared to a wide variety of deep NN based models (CNNs, RNNs, hierarchical NNs, and Transformers based models).

**To R4:** 1) Regardless of which MCMC method is used, the need to perform a sampling based iterative procedure (e.g., hundreds of MCMC iterations) for each test document limits the efficiency for out-of-sample prediction. In addition, if restricting to Gibbs sampling, it is difficult to incorporate label information into

| Methods | Accuracy | | | | Testing time in seconds | | | |
|---|---|---|---|---|---|---|---|---|
| | Reuters | ELEC | IMDB-2 | IMDB-10 | Reuters | ELEC | IMDB-2 | IMDB-10 |
| bi-conv-PGDS (TLASGR-MCMC) | 78.0 ± 0.7 | 84.5 ± 0.8 | 84.0 ± 0.8 | 37.9 ± 0.4 | 17.35 | 15.07 | 25.61 | 27.52 |
| bi-conv-PGDS (hybrid SG-MCMC and VI) | 76.8 ± 1.0 | 83.7 ± 1.2 | 82.6 ± 1.1 | 36.9 ± 0.7 | 0.15 | 0.14 | 0.17 | 0.20 |

Table 2: Comparison of the testing times (seconds) with batch-size 25.

the model. Thus, we develop an encoder network to map the observations directly to their latent representations. We also introduce a hybrid SG-MCMC/VI for inference. While [15] has validated hybrid SG-MCMC/VI empirically, we acknowledge there is still theoretical gap to fill to validate the practice of sampling from a variational posterior in lieu of the exact conditional posterior, and rolling these approximate samples into a Markov chain. This presents an interesting theoretical question (including analysis of convergence and mixing), which, however, is beyond the scope of this paper. The reason why we develop a parallelized Gibbs sampler as well as a hybrid SG-MCMC/VI is that both of them have their own advantages. The use of encoder makes our model fast in testing time, but leads to a tradeoff in accuracy, as shown in Table 2. In addition, the encoder network enables our model to directly incorporate side information.

2) Note in each topic, the words assigned with negligible weights are not important, as shown in Fig. 2; the weights of these noted meaningless phrases are: "lap ($2e^{-4}$) guess ($9e^{-3}$) none ($3e^{-4}$)", "rarely ($2e^{-4}$) though ($3e^{-4}$) packaged ($2e^{-4}$)", "pleased (0.35) roll ($2e^{-4}$) recorded ($2e^{-4}$)". 3) The validation set is not used by our models to select parameters for unsupervised learning (see discussion at the very beginning); it is used to select the step size in supervised learning. We use Adam to update the encoder of our model and use ELBO as the convergence criteria. All code can be found in corresponding papers. 4) We will add the standard deviations in Fig. 1. 5) CPGBN is not a multilayer convolutional model, but a coupling of CPFA and GBN via a probabilistic document-level pooling layer. It extends CPFA to capture the hierarchical relationships of different phrases. Comparing with

Figure 2: Example topics.

CPGBN, the proposed CPGDS focuses on the structural improvement at the sentence level by capturing the relationships of different sentences. They are two complementary ideas. In addition, comparing with CPGBN, our model has greater advantages in multi-category data, like IMDB-10 in Table 1 and yelp14 in Table 2, which are multi-level sentiment classification problems that need to consider the relationships between sentences. 6) We list more attention visualization of different datasets in Figs. 6 and 7, and they are not "cherry picked" examples; we note similar visualizations can be found in [28]. 7) We will provide more clear and simplified notation in our revision. 8) Gamma($a, 1/b$) in our paper have mean $a/b$. 9) To exploit a rich set of tools developed for count data analysis, we first link sequential binary vectors to sequential count vectors via the Bernoulli-Poisson link. This can be seen as an auxiliary variable trick to arrive at a Poisson-gamma structure that is amenable to posterior inference. 10) To utilize the reparameterization trick motivates the choice of Weibull distribution, which exhibits a similar probability density function as the gamma distribution that is not reparameterizable (see [15] for more details). 11) We will correct Line 169 and use $\lambda$ to replace $\xi$ in line 198.

[Meta-Review · NeurIPS 2020]

The paper proposes a novel hierarchical Bayesian model for text data that can capture both intra-sequence structure with convolution and inter-sentence dependency via a gamma dynamic model. The paper is well written and the authors provided extensive comparisons to other methods, showing the usefulness of the approach.